# Hamiltonian Dynamics with Non-Newtonian Momentum for Rapid Sampling

**Greg Ver Steeg**      **Aram Galstyan**
University of Southern California, Information Sciences Institute
`{gregv,galstyan}@isi.edu`

## Abstract

Sampling from an unnormalized probability distribution is a fundamental problem in machine learning with applications including Bayesian modeling, latent factor inference, and energy-based model training. After decades of research, variations of MCMC remain the default approach to sampling despite slow convergence. Auxiliary neural models can learn to speed up MCMC, but the overhead for training the extra model can be prohibitive. We propose a fundamentally different approach to this problem via a new Hamiltonian dynamics with a non-Newtonian momentum. In contrast to MCMC approaches like Hamiltonian Monte Carlo, no stochastic step is required. Instead, the proposed deterministic dynamics in an extended state space exactly sample the target distribution, specified by an energy function, under an assumption of ergodicity. Alternatively, the dynamics can be interpreted as a normalizing flow that samples a specified energy model without training. The proposed Energy Sampling Hamiltonian (ESH) dynamics have a simple form that can be solved with existing ODE solvers, but we derive a specialized solver that exhibits much better performance. ESH dynamics converge faster than their MCMC competitors enabling faster, more stable training of neural network energy models.

## 1 Introduction

While probabilistic reasoning is crucial for science and cognition [1], distributions that can be directly sampled are rare. Without sampling it is difficult to measure likely outcomes, especially in high dimensional spaces. A general purpose method to sample any target distribution, the Metropolis algorithm for Markov Chain Monte Carlo (MCMC), is considered one of the top algorithms of the 20th century [2]. The "Monte Carlo" in MCMC refers to the essential role of stochasticity, or chance, in the approach: start with any state, propose a random update, and accept it according to a weighted coin flip.

The most widely used methods for sampling mainly differ in their choice of proposals for the next state of the Markov chain. The most prominent example, Hamiltonian Monte Carlo (HMC), uses Hamiltonian dynamics to suggest proposals that quickly move over large distances in the target space [3]. HMC using the No U-

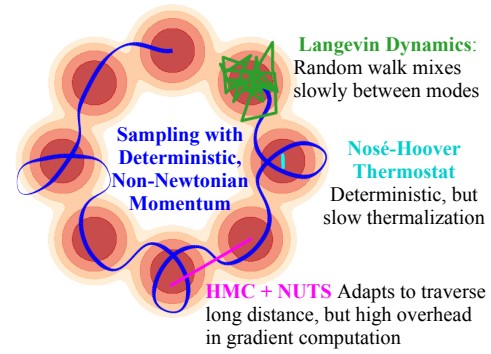

Figure 1: Sampling with 50 gradient evaluations per method. A special property of our proposed deterministic dynamics using a non-Newtonian momentum, proportional to line thickness, is that it ergodically samples from the target distribution.

35th Conference on Neural Information Processing Systems (NeurIPS 2021).

Turn Sampling (NUTS) heuristic for choosing hyper-parameters [4] forms the backbone of modern probabilistic programming languages like Stan [5]. In the limit of one discrete HMC step per proposal, we recover the widely used discrete Langevin dynamics [3; 6]. Many approaches train neural networks to improve sampling either through better proposal distributions or flexible variational distributions [7; 8; 9; 10; 11; 12; 13; 14], but these can be expensive if, for instance, the target distribution to be sampled is itself being optimized.

We propose a new Hamiltonian dynamics, based on the introduction of a non-Newtonian momentum, which leads to a deterministic dynamics that directly samples from a target energy function. The idea that deterministic dynamics in an extended space can directly sample from a Gibbs-Boltzmann distribution goes back to Nosé [15]. This principle is widely used in molecular dynamic (MD) simulations [16] but has had limited impact in machine learning. One reason is that direct application of techniques developed for MD often exhibit slow mixing in machine learning applications.

Our contribution stems from the recognition that many of the design considerations for molecular dynamics samplers are to enforce physical constraints that are irrelevant for machine learning problems. By returning to first principles and discarding those constraints, we discover a Hamiltonian whose dynamics *rapidly* sample from a target energy function by using a non-physical form for the momentum. This is in contrast to the more physical, but slower, alternative of introducing auxiliary "thermostat" variables that act as a thermal bath [17; 18; 19; 20]. The result is a fast and deterministic drop-in replacement for the MCMC sampling methods used in a wide range of machine learning applications. Alternatively, our method can be viewed as a normalizing flow in an extended state space which directly samples from an unnormalized target density without additional training.

The most significant impact of our approach is for applications where minimizing memory and computation is the priority. For training energy-based models, for example, sampling appears in the inner training loop. Re-training a neural sampler after each model update is costly and entails complex design and hyper-parameter choices. While MCMC sidesteps the training of an extra model, MCMC converges slowly and has led to widespread use of a number of dubious heuristics to reduce computation [21; 22]. Because our energy sampling dynamics does not have the stochastic, random walk component of MCMC, it converges towards low energy states much faster, especially in the transient regime, and explores modes faster as shown in Fig. 1 and demonstrated in experiments.

## 2 Energy Sampling Hamiltonian (ESH) Dynamics

We define a separable Hamiltonian, $H(\mathbf{x}, \mathbf{v})$, over position $\mathbf{x} \in \mathbb{R}^d$ and velocity $\mathbf{v} \in \mathbb{R}^d$ (we use momentum and velocity interchangeably since we set mass to 1).

$$H(\mathbf{x}, \mathbf{v}) = E(\mathbf{x}) + K(\mathbf{v})$$

The potential energy function, $E(\mathbf{x})$, is the energy for the target distribution with unknown normalization, $Z$, that we would like to sample, $p(\mathbf{x}) = e^{-E(\mathbf{x})}/Z$, and is defined by our problem. We consider an unusual form for the kinetic energy, $K(\mathbf{v})$, where $v^2 \equiv \mathbf{v} \cdot \mathbf{v}$.

$$K_{ESH}(\mathbf{v}) = d/2 \log(v^2/d) \tag{1}$$

For contrast, HMC uses Newtonian kinetic energy, $K(\mathbf{v}) = 1/2\ v^2$. Hamiltonian dynamics are defined using dot notation for time derivatives and using $\mathbf{g}(\mathbf{x}) \equiv \partial_{\mathbf{x}} E(\mathbf{x})$. We also suppress the time

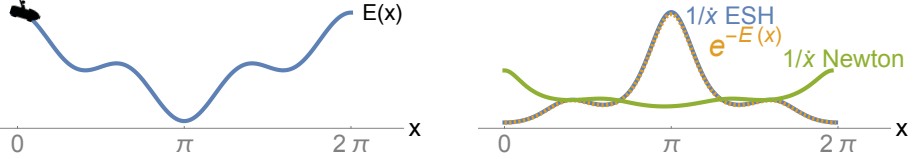

Figure 2: For Newtonian dynamics, the roller coaster spends most of its time in high energy states (left). For ESH dynamics, the time spent in the region $(x, x+dx)$ is $1/\dot{x}$ which is exactly proportional to the Boltzmann sampling probability, $e^{-E(x)}$ (right plot). Thus, sampling the ESH dynamics over time is equivalent to sampling from the target distribution.

dependence, $\mathbf{x}(t), \mathbf{v}(t)$, unless needed.

|  | General Hamiltonian | ESH Dynamics | HMC/Newtonian Dynamics |  |
|---|---|---|---|---|
|  | $\dot{\mathbf{x}} = \partial_\mathbf{v} H(\mathbf{x}, \mathbf{v})$ | $\dot{\mathbf{x}} = \mathbf{v}/(v^2/d)$ | $\dot{\mathbf{x}} = \mathbf{v}$ | (2) |
|  | $\dot{\mathbf{v}} = -\partial_\mathbf{x} H(\mathbf{x}, \mathbf{v})$ | $\dot{\mathbf{v}} = -\mathbf{g}(\mathbf{x})$ | $\dot{\mathbf{v}} = -\mathbf{g}(\mathbf{x})$ |  |

Hamiltonian dynamics have a number of useful properties. Most importantly, the Hamiltonian is conserved under the dynamics. Secondly, via Liouville's theorem we see that dynamics are volume preserving (or more generally, *symplectic*). Finally, the dynamics are reversible. While the difference in form between ESH dynamics and Newtonian dynamics appears slight, the effect is significant as we show with an example.

Imagine riding a roller-coaster running infinitely around a circular track shown in Fig. 2. $E(x)$ is the potential energy from gravity at a point in the loop specified by a scalar angle $x \in [0, 2\pi]$. At the bottom, potential energy is small while kinetic energy is large. Under Newtonian physics the speed is slowest at the top so most of a rider's time is spent in dreadful anticipation of the fall, before quickly whipping through the bottom. In both cases, the "momentum" is largest at the bottom, but for the ESH dynamics we see a different outcome. Changes in the roller-coaster position, $\dot{x}$, happen in slow motion at the bottom when momentum is large and then go into fast-forward at the top of the track where the momentum is small. A spectator will see the rider spending most of their time at the bottom, with low potential energy, and the time spent in each region is exactly proportional to the Boltzmann distribution, as we show mathematically in the next section.

An intriguing connection between Newtonian and ESH dynamics is discussed in Appendix A.1, which mirrors a thermodynamic effect emerging from Tsallis statistics that leads to anomalous diffusion [23]. Note that $v^2 = 0$ is a singularity in the dynamics in Eq. 2. For 1-D dynamics, this could be an issue, as the velocity can never change signs. However, in higher dimensions, dynamics always avoid the singularity (Appendix A.2).

## 2.1 Ergodic ESH Dynamics Sample the Target Distribution

Assume that we have an unnormalized target (or Gibbs) distribution specified by energy $E(\mathbf{x})$ as $p(\mathbf{x}) = e^{-E(\mathbf{x})}/Z$ with an unknown (but finite) normalization constant, $Z$. The reason for defining the non-standard Hamiltonian dynamics in Eq. 2 is that the uniform distribution over all states with fixed energy, $c$, gives us samples from the Gibbs distribution after marginalizing out the velocity.

$$p(\mathbf{x}, \mathbf{v}) = \delta\big(E(\mathbf{x}) + K_{ESH}(\mathbf{v}) - c\big)/Z' \qquad \rightarrow \qquad p(\mathbf{x}) = e^{-E(\mathbf{x})}/Z \qquad (3)$$

The proof is as follows.

$$\begin{aligned}
p(\mathbf{x}) &= \int d\mathbf{v}\, p(\mathbf{x}, \mathbf{v}) = 1/Z' \int d\mathbf{v}\, \delta\left(E(\mathbf{x}) + d/2 \log v^2/d - c\right) \\
&= 1/Z' \int J(\boldsymbol{\phi})\, d\boldsymbol{\phi}\, \rho^{d-1}\, d\rho\, \delta\left(E(\mathbf{x}) + d \log \rho - c - d/2 \log d\right) \\
&= 1/Z' \int J(\boldsymbol{\phi})\, d\boldsymbol{\phi}\, \rho^{d-1}\, d\rho\, \rho/d\, \delta(\rho - e^{-E(\mathbf{x})/d + c/d + 1/2 \log d}) = e^{-E(\mathbf{x})}/Z \qquad \square
\end{aligned}$$

In the second line, we switch to hyper-spherical coordinates, with $\rho = |\mathbf{v}|$ and angles combined in $\boldsymbol{\phi}$. In the third line, we use an identity for the Dirac delta that for any smooth function, $h(z)$, with one simple root, $z^*$, we have $\delta(h(z)) = \delta(z - z^*)/|h'(z^*)|$ [24]. Finally we integrate over $\rho$ using the Dirac delta, with the $\phi$ integral contributing constants absorbed into the normalizing constant.

This proof resembles results from molecular dynamics [15], where an auxiliary scalar "thermostat" variable is integrated over to recover a "canonical ensemble" in both the coordinates and momentum. In our case, the momentum variables are already auxiliary so we directly use the velocity variables in a role usually filled by the thermostat. The form of the energy is inspired by the log-oscillator [25]. Although log-oscillator thermostats never gained traction in molecular dynamics because they violate several physical properties [26], this is not relevant for machine learning applications.

Although Eq. 3 gives us the target distribution as the marginal of a uniform distribution over constant energy states, this is not helpful at first glance because it is not obvious how to sample the uniform

distribution either. At this point, we invoke the ergodic hypothesis: the dynamics will equally occupy all states of fixed energy over time. More formally, the (*phase space*) average over accessible states of equal energy is equal to the *time* average under Hamiltonian dynamics for sufficiently long times.

$$\text{Ergodic hypothesis:} \quad \frac{1}{Z'} \int d\mathbf{x} d\mathbf{v} \, \delta(H(\mathbf{x}, \mathbf{v}) - c) \, h(\mathbf{x}, \mathbf{v}) = \lim_{T \to \infty} \frac{1}{T} \int_0^T dt \, h(\mathbf{x}(t), \mathbf{v}(t)) \quad (4)$$

The importance and frequent appearance of ergodicity in physical and mathematical systems has spurred an entire field of study [27] with celebrated results by Birkhoff and von Neumann [28; 29; 30] explaining why ergodicity typically holds. Ergodicity for a specific system depends on the details of each system through $E(\mathbf{x})$, and the appearance of hidden invariants of the system breaks ergodicity [19; 31; 24]. While we do not expect hidden invariants to emerge in highly nonlinear energy functions specified by neural networks, systems can be empirically probed through fixed point analysis [20] or Lyapunov exponents [32]. Standard MCMC also assumes ergodicity, but the use of stochastic steps typically suffices to ensure ergodicity [3].

The key distinction to MCMC is that our scheme is fully deterministic, and therefore we use ergodic dynamics, rather than randomness in update proposals, to ensure that the sampler explores the entire space. As an alternative to relying on ergodicity, we also give a normalizing flow interpretation.

## 2.2 Numerical Integration of the Energy Sampling Hamiltonian (ESH) ODE

The leapfrog or Stormer-Verlet integrator [33] is the method of choice for numerically integrating standard Hamiltonian dynamics because the discrete dynamics are explicitly reversible and volume preserving. Additionally, the error of these integrators are $O(\epsilon^3)$ for numerical step size, $\epsilon$. Volume preservation is not guaranteed by off-the-shelf integrators like Runge-Kutta [18]. The analogue of the leapfrog integrator for our proposed dynamics in Eq. 2 follows.

$$\mathbf{v}(t + \epsilon/2) = \mathbf{v}(t) - \epsilon/2 \, \mathbf{g}(\mathbf{x}(t)) \qquad \text{Newtonian/HMC} \qquad \text{ESH}$$
$$\mathbf{x}(t + \epsilon) = \mathbf{x}(t) + \epsilon \, s \, \mathbf{v}(t + \epsilon/2) \qquad\qquad s = 1 \qquad s = d/v^2(t + \epsilon/2)$$
$$\mathbf{v}(t + \epsilon) = \mathbf{v}(t + \epsilon/2) - \epsilon/2 \, \mathbf{g}(\mathbf{x}(t + \epsilon)) \qquad\qquad (5)$$

Unfortunately, for ESH the effective step size for $\mathbf{x}$ can vary dramatically depending on the magnitude of the velocity. This leads to very slow integration with a fixed step size, as we will illustrate in the results. We consider two solutions to this problem: adaptive step-size Runge-Kutta integrators and a leapfrog integrator in transformed coordinates. Although Runge-Kutta integrators do not give the same guarantees in terms of approximate conservation of the Hamiltonian as symplectic integrators, the Hamiltonian was stable in experiments (App. E.1). Moreover, looking at Eq. 3, we see that the exact value of the Hamiltonian is irrelevant, so results may be less sensitive to fluctuations.

**Transformed ESH ODE** The adaptive time-step in the Runge-Kutta ODE solver is strongly correlated to the magnitude of the velocity (App. E.1). This suggests that we might be able to make the integration more efficient if we chose a time-rescaling such that the optimal step size was closer to a constant. We chose a new time variable, $\bar{t}$, so that $dt = d\bar{t} \, |\mathbf{v}|/d$, leading to the transformed dynamics $\dot{\mathbf{x}} = \mathbf{v}/|\mathbf{v}|, \dot{\mathbf{v}} = -|\mathbf{v}|/d \, \mathbf{g}(\mathbf{x})$ (for notational convenience, we omit the bar over $t$ and continue using dot notation to indicate derivative with respect to the scaled time). Next, we re-define the variables as follows: $\mathbf{u} \equiv \mathbf{v}/|\mathbf{v}|, r \equiv \log|\mathbf{v}|$. The transformed ODE follows.

$$\text{ESH Dynamics} \qquad d\bar{t} \equiv dt \, d/|\mathbf{v}| \qquad\qquad \dot{\mathbf{x}} = \mathbf{u} \qquad\qquad (6)$$
$$\dot{\mathbf{x}} = \mathbf{v}/(v^2/d) \qquad \text{with} \qquad \mathbf{u} \equiv \mathbf{v}/|\mathbf{v}| \qquad \implies \qquad \dot{\mathbf{u}} = -(\mathbb{I} - \mathbf{u}\mathbf{u}^T)\mathbf{g}(\mathbf{x})/d$$
$$\dot{\mathbf{v}} = -\mathbf{g}(\mathbf{x}) \qquad\qquad r \equiv \log|\mathbf{v}| \qquad\qquad \dot{r} = -\mathbf{u} \cdot \mathbf{g}(\mathbf{x})/d$$

Interestingly, the previously troublesome magnitude of the velocity, captured by $r$, plays no role in the dynamics of $\mathbf{x}$. However, we must still solve for $r$ because at the end of the integration, we need to re-scale the time coordinates back so that the time average in Eq. 4 can be applied. Again, we can solve this ODE with general-purpose methods like Runge-Kutta or with a leapfrog integrator.

**Leapfrog integrator for the time-scaled ESH ODE dynamics** Correctly deriving a leapfrog integrator for the time-scaled ODE is nontrivial, but turns out to be well worth the effort. The updates for $\mathbf{x}, \mathbf{u}$ are below with the full derivation including $r$ update in App. B.1.

$$\mathbf{u}(t + \epsilon/2) = \mathbf{f}(\epsilon/2, \mathbf{g}(\mathbf{x}(t)), \mathbf{u}(t)) \qquad\qquad \text{Half step in } \mathbf{u}$$
$$\mathbf{x}(t + \epsilon) = \mathbf{x}(t) + \epsilon \, \mathbf{u}(t + \epsilon/2) \qquad\qquad \text{Full step in } \mathbf{x} \qquad (7)$$
$$\mathbf{u}(t + \epsilon) = \mathbf{f}(\epsilon/2, \mathbf{g}(\mathbf{x}(t + \epsilon)), \mathbf{u}(t + \epsilon/2)) \qquad \text{Half step in } \mathbf{u}$$

$$\text{with} \quad \mathbf{f}(\epsilon, \mathbf{g}, \mathbf{u}) \equiv \frac{\mathbf{u} + \mathbf{e}\left(\sinh\left(\epsilon\,|\mathbf{g}|/d\right) + \mathbf{u}\cdot\mathbf{e}\cosh\left(\epsilon\,|\mathbf{g}|/d\right) - \mathbf{u}\cdot\mathbf{e}\right)}{\cosh\left(\epsilon\,|\mathbf{g}|/d\right) + \mathbf{u}\cdot\mathbf{e}\sinh\left(\epsilon\,|\mathbf{g}|/d\right)} \quad \text{and} \quad \mathbf{e} \equiv -\mathbf{g}/|\mathbf{g}|$$

The update for $\mathbf{u}$ is towards the direction of gradient descent, $\mathbf{e}$. If the norm of the gradient is large $\mathbf{u} \to \mathbf{e}$ and the dynamics are like gradient descent. The update form keeps $\mathbf{u}$ a unit vector.

**Ergodic sampling with the transformed ESH ODE** The solution we get from iterating Eq. 7 gives us $\mathbf{x}(\bar{t}), \mathbf{u}(\bar{t}), r(\bar{t})$ at discrete steps for the scaled time variable, $\bar{t} = 0, \epsilon, 2\epsilon, \ldots, \bar{T}$, where we re-introduce the bar notation to distinguish scaled and un-scaled solutions. We can recover the original time-scale by numerically integrating the scaling relation, $dt = d\bar{t}\,|\mathbf{v}(\bar{t})|/d$ to get $t(\bar{t}) = \int_0^{\bar{t}} d\bar{t}'|\mathbf{v}(\bar{t}')|/d = \int_0^{\bar{t}} d\bar{t}' e^{r(\bar{t}')}/d$. Using this expression, we transform our trajectory in the scaled time coordinates to points in the un-scaled coordinates, $(t, \mathbf{x}(t), \mathbf{v}(t))$, except that these values are sampled irregularly in $t$. For an arbitrary test function, $h(\mathbf{x})$, sampling works by relating target expectations (left) to trajectory expectations (right).

$$\mathbb{E}_{\mathbf{x} \sim e^{-E(\mathbf{x})}/Z}[h(\mathbf{x})] \overset{\text{Eq. 3}}{=} \mathbb{E}_{\mathbf{x} \sim p(\mathbf{x}, \mathbf{v})}[h(\mathbf{x})] \overset{\text{Eq. 4}}{\approx} \mathbb{E}_{t \sim \mathcal{U}[0,T]}[h(\mathbf{x}(t))] = \mathbb{E}_{\bar{t} \sim \mathcal{U}[0,\bar{T}]}[\mathbf{v}(\bar{t})/D\,h(\mathbf{x}(\bar{t}))]$$

The approximation is from using a finite $T$ rather than the large $T$ limit. The penultimate form justifies one procedure for ergodic sampling — we uniformly randomly choose $t \sim \mathcal{U}[0,T]$, then take $\mathbf{x}(t)$ as our sample. Because the time re-scaling might not return a sample that falls exactly at $t$, we have to interpolate between grid points to find $\mathbf{x}(t)$. Alternatively, the last expression is a weighted sample (by $|\mathbf{v}|$) in the time-scaled coordinates. We can avoid storing the whole trajectory and then sampling at the end with reservoir sampling [34]. See App. D for details.

## 2.3 Alternative Interpretation: Jarzynski Sampling with ESH as a Normalizing Flow

We can avoid the assumption of ergodicity by interpreting the ESH dynamics as a normalizing flow. The idea of interpreting Hamiltonian dynamics as a normalizing flow that can be weighted to sample a target distribution goes back to Radford Neal's unpublished [35] but influential [36; 12; 37] work on Hamiltonian Importance Sampling. Jarzynski's equality [38; 39] can be related to importance sampling [40] and has been applied to Hamiltonian dynamics [39; 24] which motivates our approach. Neal's Hamiltonian importance sampling required an annealing schedule to move the initial high energy Newtonian dynamics closer to the low energy target distribution. This is not required for ESH dynamics as our Hamiltonian directly samples the target.

We initialize our normalizing flow as $\mathbf{x}(0), \mathbf{v}(0) \sim q_0(\mathbf{x}, \mathbf{v}) = e^{-E_0(\mathbf{x})}/Z_0\,\delta(|\mathbf{v}| - 1)/A_d$, where $E_0$ is taken to be a simple energy model to sample with a tractable partition function, $Z_0$, like a unit normal. Then we transform the distribution using deterministic, invertible ESH dynamics to $q_t(\mathbf{x}(t), \mathbf{v}(t))$. The change of variables formula for the ODE in Eq. 6 (App. C) is

$$\log q_t(\mathbf{x}(t), \mathbf{v}(t)) = \log q_0(\mathbf{x}(0), \mathbf{v}(0)) + \log|\mathbf{v}(0)| - \log|\mathbf{v}(t)|.$$

Then, using importance sampling, we can relate $q_t$ to the target distribution $p(\mathbf{x}) = e^{-E(\mathbf{x})}/Z$.

$$\mathbb{E}_{q_t(\mathbf{x}(t), \mathbf{v}(t))}[e^{w(\mathbf{x}(t), \mathbf{v}(t))}\,h(\mathbf{x}(t))] = \frac{Z}{Z_0}\mathbb{E}_{p(\mathbf{x})}[h(\mathbf{x})] \tag{8}$$

$$w(\mathbf{x}(t), \mathbf{v}(t)) \equiv E_0(\mathbf{x}(0)) - E(\mathbf{x}(0)) + \log|\mathbf{v}(t)| - \log|\mathbf{v}(0)|$$

For the weights, we should formally interpret $\mathbf{x}(0), \mathbf{v}(0)$ as functions of $\mathbf{x}(t), \mathbf{v}(t)$ under the inverse dynamics. This expression holds for any $h$, including $h(\mathbf{x}) = 1$, which gives us the partition function ratio $\mathbb{E}_{q_t}[e^w] = Z/Z_0$. We use this relation to replace $Z/Z_0$ when calculating expectations, which is known as self-normalized importance sampling. Note that as in ergodic sampling in the time-scaled coordinates, the weight is proportional to $|\mathbf{v}(t)|$. We give a full derivation and tests using ESH-Jarzynski flows to estimate partition functions and train EBMs in App. C, but primarily focus on ergodic sampling results in the rest of the paper.

## 3 Results

ESH dynamics provide a fundamentally different way to sample distributions, so we would like experiments to build intuition about how the approach compares to other sampling methods, before considering applications like sampling neural network energy functions. We introduce some standard synthetic benchmarks along with variations to help contrast methods.

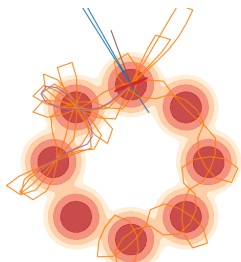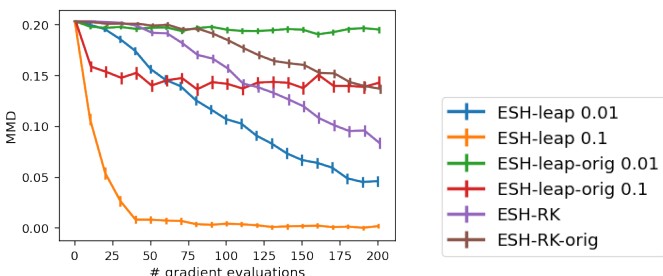

Figure 3: (Left) A single chain integrated with 200 gradient evaluations with various ESH ODE solvers on the 2D MOG energy function. (Right) Maximum Mean Discrepancy (MMD) as a function of the number of gradient evaluations. Solutions to the original, unscaled ODE are labeled `orig`.

- *Mixture of Gaussian* (2D MOG) is a mixture of 8 Gaussians with separated modes.
- *Mixture starting with informative prior* (2D MOG-prior) is the same as 2D MOG but initializes samplers from a mode of the distribution. This simulates the effect of using "informative priors"[22] to initialize MCMC chains, as in persistent contrastive divergence (PCD) [41]. The danger of informative initialization is that samplers can get stuck and miss other modes.
- *Ill Conditioned Gaussian* (50D ICG) from [3] has different length scales in different dimensions.
- *Strongly Correlated Gaussian* (2D SCG) with a Pearson correlation of $0.99$.
- *Strongly Correlated Gaussian with bias* (2D SCG-bias) is the same as 2D-SCG except we bias the initialization toward one end of the long narrow distribution (as in Fig. 5). This tests the ability of samplers to navigate long, low energy chasms.
- *Funnel* (20D Funnel) A challenging test case where the length scale in one region is exponentially smaller than other regions. Our implementation is from [10].

**Metrics**    Our goal is to obtain high quality samples from the target distribution. With the rise of multi-core CPUs, GPUs, and cloud computing, most practitioners prefer to simulate many Markov chains in parallel for the minimum amount of time, rather than to simulate one chain for a very long time [42; 21]. Therefore, we imagine running a number of samplers in parallel, and then we measure how well they resemble the target distribution after some time, as a function of *number of serial gradient evaluations* per chain. For neural network energy functions, gradient evaluation dominates the computational cost. To measure how well the samples resemble true samples from the distribution we use the (unbiased) kernel estimator of the squared **Maximum Mean Discrepancy (MMD)** [43], with Gaussian kernel with bandwidth chosen via the median heuristic. MMD is easy to compute with minimal assumptions, converges to zero if the sampling distributions are indistinguishable, and has high statistical power. In line with previous work, we also calculate the popular **Effective Sample Size (ESS)** using the method in [4]. We caution readers that ESS is a measure of the variance of our estimate *assuming* we have sampled the Markov chain long enough to get unbiased samples. For this reason, many MCMC samplers throw away thousands of initial steps in a "burn-in" period. While we perform our experiments without burn-in in the transient regime, ESS can be interpreted as a proxy for mixing speed, as it depends on the auto-correlation in a chain. Cyclic dynamics would violate ergodicity and lead to high auto-correlation and a low ESS, so this can be understood as evidence for ergodicity. Additional empirical evidence of ergodicity is given in Sec. E.2.

### 3.1 Comparing ESH Integrators

For ESH dynamics, we compare a variety of methods to solve the dynamics. First of all, we consider solving the original dynamics (`orig`), Eq. 2, versus the time-scaled dynamics, Eq. 6. For each dynamics, we compare using an adaptive Runge-Kutta (`RK`) solver [44] (fifth order Dormand-Prince [45]) to the leapfrog solvers (`leap`) in Eq. 5 and 7 respectively. In Fig. 3 the scaled dynamics are preferable to the original and the leapfrog integrator is preferable to Runge-Kutta. Sec. E.1 confirms that the leapfrog integrator for the scaled ODE is the best approach across datasets.

For leapfrog integrators, there is only one hyper-parameter, the step size, and for Runge-Kutta integrators we need to choose the tolerance. In experiments we set these to be as large as possible

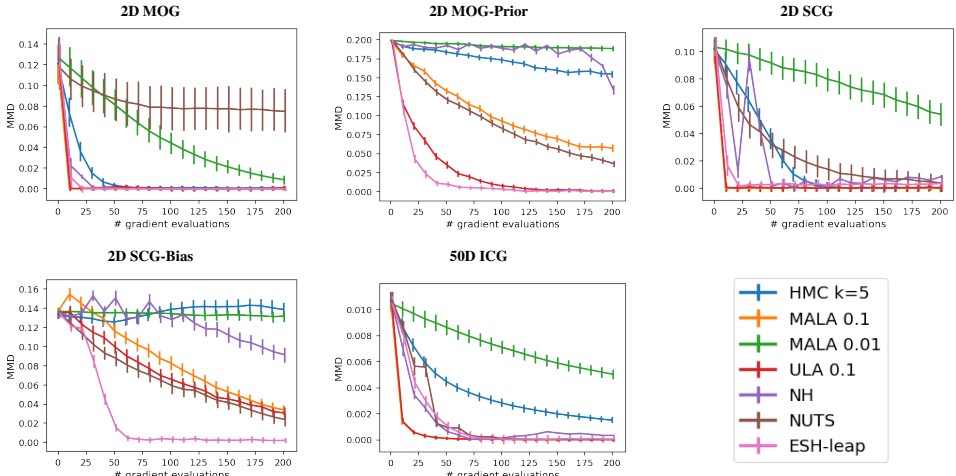

Figure 4: Maximum Mean Discrepancy (MMD) as a function of the number of gradient evaluations performed for different samplers.

on a logarithmic grid without leading to numerical errors. This leads us to use a value of $\epsilon = 0.1$ in subsequent experiments. For the Runge-Kutta integrators we were forced to use rather small sizes for the relative tolerance ($10^{-5}$) and absolute tolerance ($10^{-6}$) to avoid numerical errors. Since the Hamiltonian should be conserved, it is a good measure of a solver's stability. We plot the Hamiltonian for different solvers in App. E.1. Monitoring the error in the Hamiltonian could potentially make an effective scheme for adaptive updating of the step size. The scaled solver makes a step of fixed distance in the input space, but for distributions with varying length scales, this may be undesirable.

## 3.2 Comparing ESH with Other Samplers

We compare with the following sampling methods in our experiments that also use gradient information to sample from energy models.

- *Metropolis-Adjusted Langevin Algorithm* (MALA) [46] is a popular sampling method that uses gradient information in the Langevin step, along with a Metropolis-Hastings rejection step to ensure convergence to the target distribution.
- *Unadjusted Langevin Algorithm* (ULA) skips the Metropolis-Hastings rejection step, with the argument that if the step size is made small enough, no rejections will be necessary [6].
- *Hamiltonian Monte Carlo* (HMC) [3] uses Hamiltonian dynamics to propose Markov steps that traverse a large distance while still having a high likelihood of being accepted. In the experiments we use $k = 5$ steps to see if it improves over MALA/ULA. If more steps are beneficial, we expect this to be discovered by automatic hyper-parameter selection using NUTS.
- *No-U-Turn Sampler* (NUTS) [4] is a version of HMC with automatic hyper-parameter selection.
- *Nosé-Hoover Thermostat* (NH) is a deterministic dynamics in an extended state space that introduces a "thermostat" variable in addition to velocity. Based on the original works of Nosé [15] and Hoover [17], we used a general numerical integration scheme proposed by Martyna et al. [20] with the specific form taken from [47].

Results are summarized in Fig. 4 and Table 3.2, with a visualization in Fig. 14. Langevin dynamics give a strong baseline. Nosé-Hoover and HMC give mixed results, depending on the energy. While automatic hyper-parameter selection with NUTS is useful in principle, the overhead in gradient computation makes this approach uncompetitive in several cases. ESH is competitive in all cases and a clear favorite in certain situations. Fig. 5 illustrates why ESH is particularly effective for the biased initialization examples. Langevin dynamics have a hard time navigating long low energy chasms quickly because of random walk behavior. On the other hand, far from deep energy wells with large gradients, Langevin makes large steps, while ESH is limited to constant length steps in the input space. For this reason, ESH slightly lags Langevin in the ill-conditioned and strongly correlated Gaussians. While we may get the best of both worlds by smartly initializing ESH chains, we investigate only the

Table 1: Effective Sample Size (with Standard Deviation)

| Sampler Dataset | **ESH-leap** (Ours) | HMC k=5 | MALA 0.1 | NH | NUTS | ULA 0.1 |
|---|---|---|---|---|---|---|
| 20D Funnel | **1.0e-03** (3.0e-04) | 9.6e-04 (1.6e-04) | 8.8e-04 (1.1e-04) | 9.4e-04 (1.5e-04) | **1.0e-03** (1.8e-04) | 8.8e-04 (1.1e-04) |
| 2D MOG | **2.1e-02** (1.6e-02) | 2.5e-03 (1.6e-03) | 4.1e-03 (1.3e-03) | 3.6e-03 (1.3e-03) | 2.2e-03 (2.8e-03) | 8.8e-03 (4.6e-03) |
| 2D MOG-prior | **2.6e-02** (1.8e-02) | 3.0e-03 (8.4e-04) | 4.2e-03 (1.5e-03) | 2.7e-03 (3.8e-04) | 4.6e-03 (2.0e-03) | 8.5e-03 (4.5e-03) |
| 2D SCG | **2.4e-02** (1.4e-02) | 7.6e-03 (4.8e-03) | 1.3e-02 (8.0e-03) | 1.0e-02 (5.7e-03) | 1.2e-02 (1.1e-02) | 1.3e-02 (8.1e-03) |
| 2D SCG-bias | **8.9e-03** (1.2e-02) | 9.6e-04 (2.1e-03) | 2.9e-03 (5.1e-03) | 1.8e-03 (3.5e-03) | 2.5e-03 (4.9e-03) | 3.7e-03 (6.0e-03) |
| 50D ICG | 1.6e-04 (2.6e-04) | 2.8e-05 (4.0e-05) | 7.4e-04 (5.0e-04) | 1.8e-04 (2.2e-04) | 1.2e-04 (1.4e-04) | **7.8e-04** (5.1e-04) |

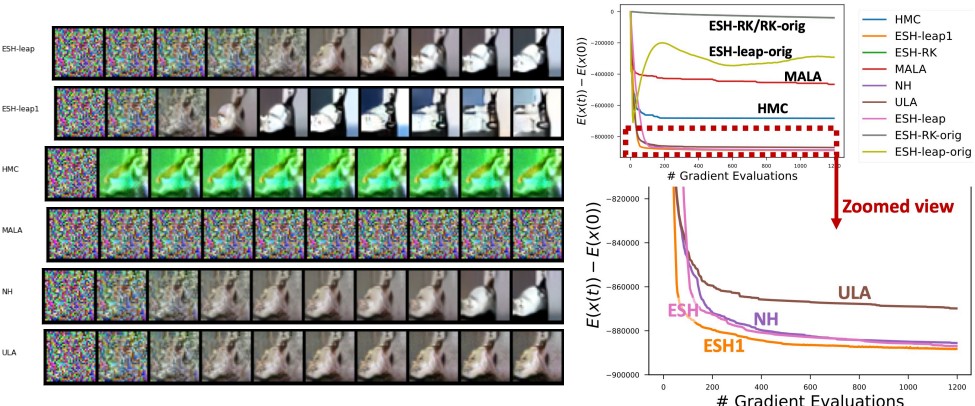

Figure 5: (Left) Navigating long energy valleys with 50 gradient evaluations. (Right) Entering deep energy wells with 5 gradient evaluations.

basic algorithm here. The large scores for effective sample size in Table 3.2 show that ESH dynamics mix quickly. All methods, including ESH, failed on the Funnel example, see App. E.3.

### 3.3 Sampling from Neural Network Energy Models

We consider sampling from a pre-trained neural network energy model, JEM [48]. Fig. 6 shows example chains for samplers starting from random initialization, and Fig. 19 shows samplers initialized from a replay buffer constructed during training. We also plot the average energy for a batch of examples. In both cases, ESH finds much lower energy samples than the algorithm used for training, ULA. Nosé-Hoover makes a surprisingly strong showing considering its poor performance on several synthetic datasets. We included an ESH leapfrog integrator with a larger step size of $\epsilon = 1$, which performed the best, especially with few gradient evaluations. HMC and MALA do poorly using noise initialization because most proposal steps are rejected. This is because JEM and similar works implicitly scale up the energy in order to take larger Langevin gradient steps that overcome random walk noise [49; 48; 22] as described in [21]. Those papers use the common argument that ULA approximates MALA for small step size [6], but with the large energy scaling employed this argument becomes dubious, as demonstrated by the large gap between ULA and MALA. The large energy scale is introduced to reduce random walk behavior, but ESH dynamics completely avoid random walks.

### 3.4 Training Neural Network Energy-Based Models

Next, we want to compare the quality of results for training neural network energy models using the current state-of-the-art methods which all use Langevin dynamics [50; 51; 52; 48; 21] to the

Figure 6: (Left) Example of sampling chains from random initialization with 200 gradient evaluations per method. (Right) Average energy over time for a batch of 50 samples using different samplers.

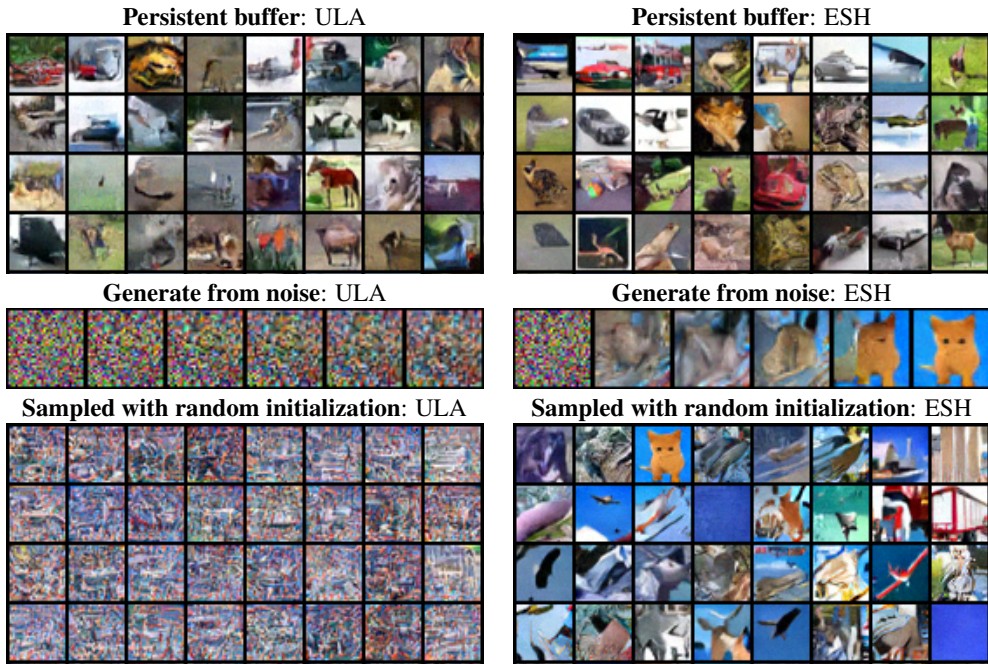

Figure 7: (Top) Samples from the persistent buffer at the end of training. (Middle) Example illustrating sampling from noise. (Bottom) Generated samples initializing from noise and using 15,000 gradient evaluations.

exact same training using ESH dynamics for sampling. In all cases, an informative prior is used for sampling, persistent contrastive divergence(PCD) [53; 41; 52]. Our synthetic results suggest ESH can be particularly beneficial with informative priors. The justification for persistent initialization in MCMC is that the target distribution is invariant under the dynamics, so if the buffer samples have converged to the target distribution then the dynamics will also produce samples from the target distribution. We show in App. B.2 that ESH dynamics also leave the target distribution invariant, justifying the use of PCD.

For our experiment, we used a setting resulting from the extensive hyper-parameter search in [21]. We train a convolutional neural network energy model on CIFAR-10 using a persistent buffer of 10,000 images. For each batch, we initialize our samplers with random momentum and image samples from the buffer, then sample using either Langevin or ESH dynamics for 100 steps, and then replace the initial image samples in the buffer. Following prior work [49] we also used spectral normalization [54] for training stability and used ensembling by averaging energy over the last ten epoch checkpoints at test time. Hyper-parameter details for training and testing are in Sec. D.4. For efficient ergodic sampling with the ESH dynamics, we do not store the entire trajectory but instead use reservoir sampling (Alg. 3).

Fig. 7 shows that the samples in the persistent buffer produced over the course of training look reasonable for both methods. However, this can be misleading and does not necessarily reflect convergence to an energy model that represents the data well [22]. Generating new samples with chains initialized from noise using 15,000 gradient evaluations per chain reveals a major difference between the learned energy models. The energy model trained with ESH dynamics produces markedly more realistic images than the one trained with Langevin dynamics.

We also tried the Jarzynski sampler for training energy-based models, with some results on toy data shown in Fig. 8. In this case, the unbiased Jarzynski sampler is very effective at learning to crisply represent boundaries with a small number of total gradient evaluations. However, for higher-dimensional data like CIFAR we found that the higher variance of the sampler becomes problematic. Training details, additional results, and discussion are in App C.3.

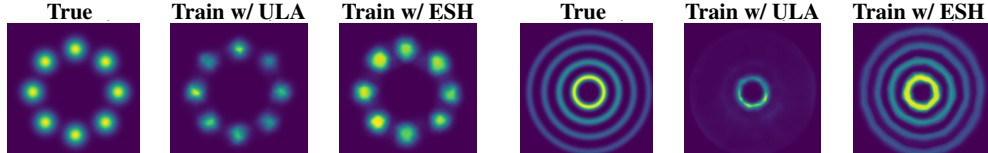

| True | Train w/ ULA | Train w/ ESH | True | Train w/ ULA | Train w/ ESH |

Figure 8: Trained neural energy models with ULA versus ESH sampling using a total of 500k gradient evaluations over the entire course of training.

## 4 Related Work

Over half a century of research on simulating Molecular Dynamics (MD) has focused on a core issue that also vexes machine learning: how to sample from an energy model. The methods employed in both fields have little overlap because MD is concerned with accurately simulating all physical properties of the target system, including momentum, dynamics of thermalization, and interaction between system and heat bath. Our experiments with Nosé-Hoover and other thermostats [15; 17] often oscillated or converged slowly because the thermostat variable is only weakly coupled to the physical, Newtonian momenta, which are then coupled to the only variables of interest in our case, the original coordinates. A recent review of MD research outlines different methods for constructing thermostats [55] including iso-kinetic thermostats [56] which have some similarities to ESH.

Ideas from physics like Hamiltonian dynamics and nonequilibrium thermodynamics have inspired many approaches in machine learning [3; 35; 36; 37; 40; 12; 57; 58; 59; 60]. Recent work also explores novel ways to combine Hamiltonian dynamics with Jarzynski's equality to derive non-equilibrium samplers [61; 62]. Another popular twist on HMC is Riemannian HMC [63], where $K(\mathbf{v}, \mathbf{x}) = \frac{1}{2}\mathbf{v}^T M(\mathbf{x})\mathbf{v}$ still represents a Newtonian momentum but in curved space. This approach requires second order information like Hessians to define the curvature, and we only considered first order approaches in this paper. Another promising line of research recognizes that many of the properties that make Hamiltonian dynamics useful for MCMC come from being part of the class of involutive operators [64] or more general orbits [65]. Another recent line of work also explores sampling via deterministic, continuous dynamics [66].

## 5 Conclusion

We presented a new approach for sampling based on deterministic, invertible dynamics and demonstrated its benefits. Removing stochasticity leads to faster mixing between modes and could enable applications where backpropagation through the sampler is desired. State-of-the-art generative models directly model the score function, the gradient of the energy function, [67] which can be used to sample with Langevin dynamics but could also potentially benefit from faster mixing with ESH dynamics. While we introduced an ergodic assumption to motivate the approach, we also provided an intuitive normalizing flow interpretation that does not require ergodicity. ESH dynamics provide a simple, fast, and deterministic drop-in replacement for sampling methods like HMC and Langevin dynamics with the potential to impact a wide range of applications such as Bayesian modeling [68] and latent factor inference [69].

### Acknowledgments and Disclosure of Funding

We thank Rob Brekelmans for helpful comments on this paper and acknowledge support from the Defense Advanced Research Projects Agency (DARPA) under award FA8750-17-C-0106. We also thank the AWS Cloud Credit for Research Program.

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
