## Supplementary Material for "Hamiltonian Dynamics with Non-Newtonian Momentum for Rapid Sampling"

## A   Properties of ESH Dynamics

### A.1   q-Hamiltonian dynamics

We can define a continuum of dynamics with Newtonian dynamics at one extreme and energy-sampling dynamics at the other via Tsallis statistics. We define a Hamiltonian over position and momentum / velocity variables (we use these terms interchangeably since we set mass to 1), $\mathbf{x}, \mathbf{v} \in \mathbb{R}^d$.

$$H(\mathbf{x}, \mathbf{v}) = E(\mathbf{x}) + K(\mathbf{v})$$

The potential energy function, $E(\mathbf{x})$, is the target distribution that we would like to sample and is defined by our problem. The kinetic energy, $K$, can be chosen in a variety of ways, but we consider the following class.

$$K_q(\mathbf{v}) = d/2 \log_q(v^2/d)$$

We use the $q$-logarithm from Tsallis statistics defined as $log_q(z) = (z^{1-q} - 1)/(1 - q)$. In the limit $q = 1$ we recover the standard logarithm and when $q = 0$ the kinetic energy simplifies to the standard, Newtonian, form used in HMC [3], $K_0(\mathbf{v}) = 1/2v^2$. Hamiltonian dynamics are defined as follows, using the dot notation for time derivatives.

$$\dot{\mathbf{x}} = \partial H/\partial \mathbf{v} \quad = \mathbf{v}/(v^2/d)^q \tag{9}$$

$$\dot{\mathbf{v}} = -\partial H/\partial \mathbf{x} = -\mathbf{g}(\mathbf{x}) \equiv -dE/d\mathbf{x} \tag{10}$$

We did not explore dynamics with intermediate values of $q$, because we only get energy-sampling dynamics for $q = 1$. However, we found it interesting that Newtonian and ESH kinetic energy terms could be viewed as opposite ends of a spectrum defined by Tsallis statistics.

The situation mathematically resembles the thermodynamic continuum that emerges from Tsallis statistics. In that case, standard thermodynamics with Brownian motion corresponds to one extreme for $q$ and in the other extreme so-called "anomalous diffusion" occurs [23]. Analogously, we can refer to the $q$-logarithmic counterpart of Newtonian momentum as anomalous momentum.

### A.2   The Zero Velocity Singularity

The equation $\dot{\mathbf{x}} = \mathbf{v}/v^2/d$ diverges when $v^2 = 0$. In 1-d, this is problematic because the sign of $v$ can not switch without going through this singularity. The example in Fig. 2 was 1-d and did give the correct ergodic distribution, but this relied on the fact that the domain was periodic so $v$ never needed to change signs. For $d > 1$, however, this singularity is not an issue. The reason is that conservation of the Hamiltonian by the dynamics prevents $v^2 = 0$. Consider a state, $\mathbf{x}(0), \mathbf{v}(0)$, with $H(\mathbf{x}(0), \mathbf{v}(0)) = c$. Under the dynamics, we can verify that $\dot{H}(\mathbf{x}(t), \mathbf{v}(t)) = 0$ so that $H(\mathbf{x}(t), \mathbf{v}(t)) = c$ for all time. Therefore, for the ESH Hamiltonian, we have $E(\mathbf{x}(t)) + d/2 \log v(t)^2 = c$. Then $v(t)^2 = e^{2/d(c-E(\mathbf{x}(t)))}$, which is positive.

## B   Derivations

### B.1   Leapfrog Derivation for Scaled Energy Sampling Hamiltonian ODE

We have the following first order ODE.

$$\dot{\mathbf{z}} = \mathcal{O}\mathbf{z} \tag{11}$$

This ODE involves the operator,

$$\mathcal{O} \equiv \mathbf{u} \cdot \partial_{\mathbf{x}} - 1/d \, \mathbf{g}(\mathbf{x})^T (\mathbb{I} - \mathbf{u}\mathbf{u}^T)\partial_{\mathbf{u}} - 1/d \, \mathbf{u} \cdot \mathbf{g}(\mathbf{x})\partial_r \text{ and } \mathbf{z} \equiv (\mathbf{x}, \mathbf{u}, r).$$

Expanding the equation shows that this equation is equivalent to Eq. 6. To be more explicit, $\mathbf{z} \equiv (x_1, \ldots, x_d, u_1, \ldots u_d, r)$ the operator $\mathcal{O}$ is a scalar operator that is applied to each element of the vector $\mathbf{z}$, so we have $2d + 1$ coupled differential equations, $\dot{z}_i = \mathcal{O}z_i$.

The solution to this ODE for small $t$ can then formally be written as $\mathbf{z}(t) = e^{t\mathcal{O}}\mathbf{z}(0)$ [18]. This can be seen by simply defining the exponential of an operator in terms of its Taylor series, then confirming this obeys Eq. 11. The leapfrog comes from the following approximation.

$$e^{\epsilon(\mathcal{O}_1+\mathcal{O}_2)} = e^{\epsilon/2\mathcal{O}_2}e^{\epsilon\mathcal{O}_1}e^{\epsilon/2\mathcal{O}_2} + O(\epsilon^3)$$

Note that we can't break up the exponent in the usual way because the operators do not commute. Instead the approximation above is derived using the Baker-Campbell-Hausdorff formula. If we set $\mathcal{O}_1 = \mathbf{u}\cdot\partial_\mathbf{x}$ then $\mathbf{x}(t+\epsilon) = e^{\epsilon\mathcal{O}_1}\mathbf{x}(t) = \mathbf{x}(t) + \epsilon\,\mathbf{u}(t)$, as in the standard leapfrog formula, Eq. 5. This can be seen by expanding the exponential as a time-series and noting that terms second order and above are zero.

To derive the updates for $\mathbf{u}, r$, we note that the solution to $z(t+\epsilon) = e^{\epsilon\mathcal{O}_2}z(t)$ is the same as the solution to the differential equation

$$\dot{\mathbf{u}} = -(\mathbb{I} - \mathbf{u}\mathbf{u}^T)\mathbf{g}/d \qquad \dot{r} = -\mathbf{u}\cdot\mathbf{g}/d \tag{12}$$

where $\mathbf{g} \equiv \mathbf{g}(\mathbf{x}(t))$ with $\mathbf{x}$ kept fixed to a constant. Finally, we just need to confirm that the proposed form of the leapfrog updates, below, satisfy this differential equation.

$$\mathbf{u}(t+\epsilon) = \frac{\mathbf{u}(t) + \mathbf{e}\,(\sinh{(\epsilon\,|\mathbf{g}|/d)} + \mathbf{u}(t)\cdot\mathbf{e}\cosh{(\epsilon\,|\mathbf{g}|/d)} - \mathbf{u}(t)\cdot\mathbf{e})}{\cosh{(\epsilon\,|\mathbf{g}|/d)} + \mathbf{u}(t)\cdot\mathbf{e}\sinh{(\epsilon\,|\mathbf{g}|/d)}} \qquad \mathbf{g} \equiv \mathbf{g}(\mathbf{x}(t)) \tag{13}$$

$$r(t+\epsilon) = r(t) + \log(\cosh{(\epsilon\,|\mathbf{g}|/d)} + \mathbf{u}\cdot\mathbf{e}\sinh{(\epsilon\,|\mathbf{g}|/d)}) \qquad\qquad \mathbf{e} \equiv -\mathbf{g}/|\mathbf{g}| \tag{14}$$

Consider solving Eq. 12 with fixed $\mathbf{g}$ for $\mathbf{u}(t)$ with initial condition $\mathbf{u}(0) = \mathbf{u}_0$. This is a vector version of the Ricatti equation. Restating the proposed form of the solution from Eq. 13 and re-arranging gives

$$\mathbf{u}(t) = \frac{\mathbf{u}_0 - \mathbf{u}_0\cdot\mathbf{e}\,\mathbf{e} + \mathbf{e}\,(\sinh{(t\,|\mathbf{g}|/d)} + \mathbf{u}_0\cdot\mathbf{e}\cosh{(t\,|\mathbf{g}|/d)})}{\cosh{(t\,|\mathbf{g}|/d)} + \mathbf{u}_0\cdot\mathbf{e}\sinh{(t\,|\mathbf{g}|/d)}}.$$

First, note that at $t = 0$, we get the correct initial condition. Taking the time derivative, we get the following.

$$\dot{\mathbf{u}} = d\mathbf{u}(t)/dt = d/dt\frac{\mathbf{u}_0 - \mathbf{u}_0\cdot\mathbf{e}\,\mathbf{e} + \mathbf{e}\,(\sinh{(t\,|\mathbf{g}|/d)} + \mathbf{u}_0\cdot\mathbf{e}\cosh{(t\,|\mathbf{g}|/d)})}{\cosh{(t\,|\mathbf{g}|/d)} + \mathbf{u}_0\cdot\mathbf{e}\sinh{(t\,|\mathbf{g}|/d)}}$$

$$= -|\mathbf{g}|/d\,\mathbf{u}(t)\frac{\sinh{(t\,|\mathbf{g}|/d)} + \mathbf{u}_0\cdot\mathbf{e}\cosh{(t\,|\mathbf{g}|/d)}}{\cosh{(t\,|\mathbf{g}|/d)} + \mathbf{u}_0\cdot\mathbf{e}\sinh{(t\,|\mathbf{g}|/d)}} + \mathbf{e}\,|\mathbf{g}|/d\,\frac{\cosh{(t\,|\mathbf{g}|/d)} + \mathbf{u}_0\cdot\mathbf{e}\sinh{(t\,|\mathbf{g}|/d)}}{\cosh{(t\,|\mathbf{g}|/d)} + \mathbf{u}_0\cdot\mathbf{e}\sinh{(t\,|\mathbf{g}|/d)}}$$

$$= \mathbf{u}(t)\,\mathbf{u}(t)\cdot\mathbf{g}/d - \mathbf{g}/d = -(\mathbb{I} - \mathbf{u}(t)\mathbf{u}(t)^T)\mathbf{g}/d \qquad \square$$

We have already used that $-\mathbf{u}(t)\cdot\mathbf{g}/d = \frac{\sinh{(t\,|\mathbf{g}|/d)}+\mathbf{u}_0\cdot\mathbf{e}\cosh{(t\,|\mathbf{g}|/d)}}{\cosh{(t\,|\mathbf{g}|/d)}+\mathbf{u}_0\cdot\mathbf{e}\sinh{(t\,|\mathbf{g}|/d)}}|\mathbf{g}|/d$. Differentiating $r(t) = \log(\cosh{(t\,|\mathbf{g}|/d)} + \mathbf{u}_0\cdot\mathbf{e}\sinh{(t\,|\mathbf{g}|/d)})$ directly recovers the correct differential equation, $\dot{r} = -\mathbf{u}\cdot\mathbf{g}/d$, with initial condition $r(0) = 0$.

The full updates including $r$ are as follows.

$$\begin{array}{lll}
r(t+\epsilon/2) = r(t) + a(\epsilon/2, \mathbf{g}(\mathbf{x}(t))) & \text{Half step in } r & \\
\mathbf{u}(t+\epsilon/2) = \mathbf{f}(\epsilon/2, \mathbf{g}(\mathbf{x}(t)), \mathbf{u}(t)) & \text{Half step in } \mathbf{u} & \\
\mathbf{x}(t+\epsilon) = \mathbf{x}(t) + \epsilon\,\mathbf{u}(t+\epsilon/2) & \text{Full step in } \mathbf{x} & (15) \\
\mathbf{u}(t+\epsilon) = \mathbf{f}(\epsilon/2, \mathbf{g}(\mathbf{x}(t+\epsilon)), \mathbf{u}(t+\epsilon/2)) & \text{Half step in } \mathbf{u} & \\
r(t+\epsilon) = r(t+\epsilon/2) + a(\epsilon/2, \mathbf{g}(\mathbf{x}(t+\epsilon))) & \text{Half step in } r &
\end{array}$$

$$\text{with}\quad \mathbf{f}(\epsilon, \mathbf{g}, \mathbf{u}) \equiv \frac{\mathbf{u} + \mathbf{e}\,(\sinh{(\epsilon\,|\mathbf{g}|/d)} + \mathbf{u}\cdot\mathbf{e}\cosh{(\epsilon\,|\mathbf{g}|/d)} - \mathbf{u}\cdot\mathbf{e})}{\cosh{(\epsilon\,|\mathbf{g}|/d)} + \mathbf{u}\cdot\mathbf{e}\sinh{(\epsilon\,|\mathbf{g}|/d)}}$$

$$a(\epsilon, \mathbf{g}) \equiv \log(\cosh{(\epsilon\,|\mathbf{g}|/d)} + \mathbf{u}\cdot\mathbf{e}\sinh{(\epsilon\,|\mathbf{g}|/d)})$$

$$\text{and using}\quad \mathbf{e} \equiv -\mathbf{g}/|\mathbf{g}|$$

As in Eq. 5, the leapfrog consists of a half-step in one set of variables $(\mathbf{u}, r)$, a full step in the other $(\mathbf{x})$, followed by another half step in the first set. This may appear to require two gradient evaluations per step. However, note that chaining leapfrogs together allows us to re-use the gradient from the previous step, requiring only one gradient evaluation per step.

## B.2 Stationary Distribution of ESH Dynamics

In the experiments, we used a persistent buffer for initialization. For MCMC, the justification is clear. If the buffer contains samples that are drawn from the target distribution then the transitions, which obey detailed balance, will produce samples that are also drawn from the target distribution. While it seems intuitive that starting with samples from the target distribution should be helpful, in this appendix we examine this claim more closely. We give an argument based on analysis of stationary distributions for why informative initialization with persistent buffers should also be useful for the scaled ESH dynamics we use in our experiments.

We would like to see how the distribution, $q(\mathbf{x}, \mathbf{v}, t)$, changes when we apply the scaled ESH dynamics, assuming that $q(\mathbf{x}, \mathbf{v}, 0) = \pi(\mathbf{x}, \mathbf{v}) \equiv e^{-E(\mathbf{x})} e^{-K(|v|)}/Z$. Here we assume that we have initialized the dynamics with samples from the target distribution and with any spherically symmetric velocities (as the magnitude of the velocity doesn't actually matter for the dynamics of $\mathbf{x}$). The scaled ESH dynamics are $\dot{\mathbf{x}} = \mathbf{v}/|v|, \dot{\mathbf{v}} = -|v|\mathbf{g}(\mathbf{x})/d$. We would like to look at how the distribution of samples changes after applying these dynamics.

$$\partial_t q(\mathbf{x}, t)|_{t=0} = \int \partial_t q(\mathbf{x}, \mathbf{v}, t) \, dv \bigg|_{t=0} = -\int (\nabla_{\mathbf{x}} \cdot (\dot{\mathbf{x}} \, \pi(\mathbf{x}, \mathbf{v})) + \nabla_{\mathbf{v}} \cdot (\dot{\mathbf{v}} \, \pi(\mathbf{x}, \mathbf{v}))) \, dv$$

In the first expression, we expand $q$ in terms of the joint distribution, and in the second we use the continuity equation. We use $dv$ as a shorthand for the volume element for $\mathbf{v}$, $\partial_t \equiv \partial/\partial t$, and $\nabla_{\mathbf{x}} \cdot \mathbf{f} \equiv \sum_i \partial_{x_i} f_i$ is the divergence operator. Replacing the time derivatives using the dynamics we get the following.

$$\begin{aligned}
\partial_t q(\mathbf{x}, t) &= -\int (\nabla_{\mathbf{x}} \cdot (\dot{\mathbf{x}} \, \pi(\mathbf{x}, \mathbf{v})) + \nabla_{\mathbf{v}} \cdot (\dot{\mathbf{v}} \, \pi(\mathbf{x}, \mathbf{v}))) \, dv \\
&= \int (\mathbf{v}/|v| \cdot \mathbf{g}(\mathbf{x}) \, \pi(\mathbf{x}, \mathbf{v}) + \pi(\mathbf{x}, \mathbf{v}) \, \nabla_{\mathbf{v}} \cdot (|v|\mathbf{g}(\mathbf{x})/d) + |v|\mathbf{g}(\mathbf{x})/d \cdot \nabla_{\mathbf{v}} \pi(\mathbf{x}, \mathbf{v})) \, dv \\
&= \int \pi(\mathbf{x}, \mathbf{v})(\mathbf{v}/|v| \cdot \mathbf{g}(\mathbf{x}) + \mathbf{g}(\mathbf{x})/d \cdot \mathbf{v}/|v| + |v|\mathbf{g}(\mathbf{x})/d \cdot (-K'(|v|)\mathbf{v}/|v|)) \, dv \\
&= \int e^{-E(\mathbf{x})} e^{-K(|v|)}/Z \, \mathbf{g}(\mathbf{x})/d \cdot \mathbf{v}/|v|(d + 1 - K'(|v|)|v|)) \, dv \\
&= 0 \quad \square
\end{aligned}$$

The first four lines go through straightforward calculations using vector calculus product and chain rules. In the fourth line, we see that the integral over $\mathbf{v}$ depends almost completely on the magnitude, $|v|$, except for a single linear (anti-symmetric) term that depends on the direction $\mathbf{v}/|v|$. We can split the integral over the direction into two parts which have the same magnitude but opposite sign, therefore canceling out to give zero.

What this calculation shows is that if we start with a sample from a buffer which is already converged to the target distribution, and initialize our velocities in a sperically symmetric way, then the density on $\mathbf{x}$ remains in the target distribution.

## B.3 Objective for Training Energy-based Models

To optimize energy-based models, $p_\theta(x) = e^{-E_\theta(x)}/Z_\theta$, we require gradients like the following.

$$\frac{d}{d\theta} \log p_\theta(x) = -\frac{dE_\theta(x)}{d\theta} - \frac{d \log Z_\theta}{d\theta} \tag{16}$$

If the energy function is specified by a neural network, the first term presents no obstacle and can be handled by automatic differentiation libraries. The negative log partition function is often called the free energy and its derivative follows.

$$-\frac{d \log Z_\theta}{d\theta} = -\frac{1}{Z_\theta} \frac{d}{d\theta} \int e^{-E_\theta(x)} dx \quad = \int \frac{e^{-E_\theta(x)}}{Z_\theta} \frac{dE_\theta(x)}{d\theta} dx = \mathbb{E}_{p_\theta(x)} \left[ \frac{dE_\theta(x)}{d\theta} \right] \tag{17}$$

For instance, if we were optimizing cross entropy between a data distribution, $p_d(x)$, and the distribution represented by our energy model, we would minimize the following loss, $\mathcal{L}$, whose

gradient $d\mathcal{L}/d\theta$ can be written using Eq. 17.

$$\mathcal{L} = -\mathbb{E}_{p_d(x)}[\log p_\theta(x)] = \mathbb{E}_{p_d(x)}[E_\theta(x)] + \log Z_\theta$$
$$\frac{d\mathcal{L}}{d\theta} = \mathbb{E}_{p_d(x)}\left[\frac{dE_\theta(x)}{d\theta}\right] - \mathbb{E}_{p_\theta(x)}\left[\frac{dE_\theta(x)}{d\theta}\right] \tag{18}$$

This objective requires us to draw (positive) samples from the data distribution and (negative) samples from the model distribution. Intuitively, the objective tries to reduce the energy of the data samples while increasing the energy of negative samples drawn from the model. When the data distribution and the energy model distribution match, the gradient will be zero.

## C   Jarzynski Sampling Derivation and Results

### C.1   Deriving the Jarzynski Sampling Relation

**ESH Dynamics as a Normalizing Flow**   We initialize our normalizing flow as $\mathbf{x}(0), \mathbf{v}(0) \sim q_0(\mathbf{x}, \mathbf{v}) = e^{-E_0(\mathbf{x})}/Z_0 \ q_0(\mathbf{v})$, where $E_0$ is taken to be a simple distribution to sample with a tractable partition function, $Z_0$, like a unit normal. The distribution $q_0(\mathbf{v}) = \delta(|\mathbf{v}| - 1)/A_d$, where $A_d$ is the area of the $d$-sphere. Because of the delta function, this is a normalizing flow on a $2d - 1$ dimensional space.

We transform the distribution using ESH dynamics to $q_t(\mathbf{x}(t), \mathbf{v}(t))$, via the deterministic and invertible transformation, $(\mathbf{x}(t), \mathbf{v}(t)) = e^{t\mathcal{O}}(\mathbf{x}(0), \mathbf{v}(0))$, or equivalently via the solution to the following ODE,

$$\dot{\mathbf{x}} = \mathbf{v}/|\mathbf{v}|, \dot{\mathbf{v}} = -|\mathbf{v}|/d \ \mathbf{g}(\mathbf{x}).$$

The distribution $q_t(\mathbf{x}(t), \mathbf{v}(t))$ is related to the original distribution via the continuous change of variables formula (or "phase space compression factor"[24]) which can be seen as a consequence of the conservation of probability and the divergence theorem or derived as the limit of the discrete change of variables formula [44].

$$d/dt \ \log q_t(\mathbf{x}(t), \mathbf{v}(t)) = {\color{red}-\partial_{\mathbf{x}} \cdot \dot{\mathbf{x}}} - \partial_{\mathbf{v}} \cdot \dot{\mathbf{v}}$$
$$= {\color{red}0} + \partial_{\mathbf{v}} \cdot (|\mathbf{v}|/d \ \mathbf{g}(\mathbf{x}))$$
$$= 1/d \ \mathbf{g}(\mathbf{x}) \cdot \mathbf{v}/|\mathbf{v}|$$
$$d/dt \ \log |\mathbf{v}| = 1/v^2 \ \mathbf{v} \cdot \dot{\mathbf{v}} = -1/d \ \mathbf{g}(\mathbf{x}) \cdot \mathbf{v}/|\mathbf{v}|$$

The last two lines give us $d/dt \ \log q_t(\mathbf{x}(t), \mathbf{v}(t)) = -d/dt \ \log |\mathbf{v}(t)|$, and therefore we get the following formula by integrating both sides.

$$\log q_t(\mathbf{x}(t), \mathbf{v}(t)) = \log q_0(\mathbf{x}(0), \mathbf{v}(0)) + \log |\mathbf{v}(0)| - \log |\mathbf{v}(t)| \tag{19}$$

This expression is the discrete change of variables for the transformation from $q_0(\mathbf{x}(0), \mathbf{v}(0))$ to $q_t(\mathbf{x}(t), \mathbf{v}(t))$, and we can now pick out the log-determinant of the Jacobian as $\log |\det \frac{\partial(\mathbf{x}(0), \mathbf{v}(0))}{\partial(\mathbf{x}(t), \mathbf{v}(t))}| = \log |\mathbf{v}(0)| - \log |\mathbf{v}(t)|$.

**Deriving a Jarzynski Equality for ESH Dynamics**   The essence of the Jarzynski equality [38], and non-equilibrium thermodynamics in general, is that distributions for dynamics that are far from equilibrium can be related to the equilibrium distribution through path weighting. Mathematically, the formulation of annealed importance sampling [57] is identical, but we prefer the broader motivation of nonequilibrium thermodynamics since our scheme does not have anything that could be viewed as an annealing schedule, and Jarzynski relations have been previously derived for continuous, deterministic dynamics [24; 39].

We begin with the target, "equilibrium", distribution that we want to estimate expectations under, $p(\mathbf{x}) = e^{-E(\mathbf{x})}/Z$. The non-equilibrium path is defined by the normalizing flow dynamics above with distribution $q_t(\mathbf{x}_t, \mathbf{v}_t)$, where for readability we will distinguish the variables at time $t$ and at

time 0 with subscripts.

$$\mathbb{E}_{p(\mathbf{x}_t)}[h(\mathbf{x}_t)] = \frac{1}{Z}\int d\mathbf{x}_t h(\mathbf{x}_t)e^{-E(\mathbf{x}_t)} = \frac{1}{Z}\int d\mathbf{x}_t d\mathbf{v}_t h(\mathbf{x}_t)e^{-E(\mathbf{x}_t)}q_t(\mathbf{v}_t|\mathbf{x}_t)$$

$$= \frac{1}{Z}\int d\mathbf{x}_t d\mathbf{v}_t h(\mathbf{x}_t)e^{-E(\mathbf{x}_0)+d\log|\mathbf{v}_t|-d\log|\mathbf{v}_0|}q_t(\mathbf{v}_t|\mathbf{x}_t) \quad \text{Conservation of Hamiltonian}$$

$$= \frac{Z_0}{Z}\int d\mathbf{x}_t d\mathbf{v}_t h(\mathbf{x}_t)e^{-E_0(\mathbf{x}_0)}/Z_0 e^{E_0(\mathbf{x}_0)-E(\mathbf{x}_0)+d\log|\mathbf{v}_t|-d\log|\mathbf{v}_0|}q_t(\mathbf{v}_t|\mathbf{x}_t)$$

$$= \frac{Z_0}{Z}\int d\mathbf{x}_t d\mathbf{v}_t h(\mathbf{x}_t)q_0(\mathbf{x}_0,\mathbf{v}_0)e^{E_0(\mathbf{x}_0)-E(\mathbf{x}_0)+d\log|\mathbf{v}_t|-d\log|\mathbf{v}_0|}\frac{q_t(\mathbf{v}_t|\mathbf{x}_t)}{q_0(\mathbf{v}_0)}$$

$$= \frac{Z_0}{Z}\int d\mathbf{x}_0 d\mathbf{v}_0 h(\mathbf{x}_t)q_0(\mathbf{x}_0,\mathbf{v}_0)e^{E_0(\mathbf{x}_0)-E(\mathbf{x}_0)+(d+1)(\log|\mathbf{v}_t|-\log|\mathbf{v}_0|)}\frac{q_t(\mathbf{v}_t|\mathbf{x}_t)}{q_0(\mathbf{v}_0)}$$

In the last line, we apply a change of variables using the determinant of the Jacobian derived for the normalizing flow. Besides multiplying by factors of 1 and re-arranging, the main step was to use the conservation of the Hamiltonian for the dynamics in the second line. The next tricky step is to find an expression for $\frac{q_t(\mathbf{v}_t|\mathbf{x}_t)}{q_0(\mathbf{v}_0)}$. First, we change the velocity variables to hyper-spherical coordinates, $\rho, \phi$, where the well known formula for the determinant of the Jacobian is $|\det\frac{\partial \mathbf{v}}{\partial(\rho,\phi)}| = \rho^{d-1}J(\phi)$.

$$\frac{q_t(\mathbf{v}_t|\mathbf{x}_t)}{q_0(\mathbf{v}_0)} = \frac{q_t(\rho_t,\phi_t|\mathbf{x}_t)}{q_0(\rho_0,\phi_0)}\frac{\rho_0^{d-1}J(\phi_0)}{\rho_t^{d-1}J(\phi_t)} \qquad \text{Change to spherical}$$

$$= \frac{q_t(\rho_t|\phi_t,\mathbf{x}_t)J(\phi_t)/A_d}{\delta(\rho_0-1)J(\phi_0)/A_d}\frac{\rho_0^{d-1}J(\phi_0)}{\rho_t^{d-1}J(\phi_t)} \qquad \text{Expand } q$$

$$= \frac{\delta(\rho_t-\rho_0 e^{(E(\mathbf{x}_0)-E(\mathbf{x}_t))/d})}{\delta(\rho_0-1)}\frac{\rho_0^{d-1}}{\rho_t^{d-1}} \qquad \text{Conservation of Hamiltonian}$$

$$= \frac{\rho_0^d}{\rho_t^d} = \frac{|\mathbf{v}_0|^d}{|\mathbf{v}_t|^d} \qquad \text{Dirac property + Cons. Hamiltonian}$$

Now we can plug this expression back into our previous derivation to get the following.

$$\mathbb{E}_{p(\mathbf{x}_t)}[h(\mathbf{x}_t)] = \frac{Z_0}{Z}\int d\mathbf{x}_0 d\mathbf{v}_0 h(\mathbf{x}_t)q_0(\mathbf{x}_0,\mathbf{v}_0)e^{E_0(\mathbf{x}_0)-E(\mathbf{x}_0)+\log|\mathbf{v}_t|-\log|\mathbf{v}_0|}$$

$$\mathbb{E}_{p(\mathbf{x}_t)}[h(\mathbf{x}_t)] = \frac{Z_0}{Z}\mathbb{E}_{q_0(\mathbf{x}_0,\mathbf{v}_0)}[e^{w(\mathbf{x}_0,\mathbf{v}_0)}h(\mathbf{x}_t)]$$

$$w(\mathbf{x}_0,\mathbf{v}_0) \equiv E_0(\mathbf{x}_0)-E(\mathbf{x}_0)+\log|\mathbf{v}_t|-\log|\mathbf{v}_0| \quad \square$$

This recovers the expression in Eq. 8. Note that because of the invertible relationship between $\mathbf{x}_t, \mathbf{v}_t$ and $\mathbf{x}_0, \mathbf{v}_0$ we can view this as an expectation over either set of variables, where the other set is considered a function of the first. We use this expression for sampling in Alg. 4.

## C.2 Testing Jarzynski ESH Sampler by Estimating the Partition Function

To verify this expression, we take a case where the ground truth partition function is known, an energy model for a random Gaussian distribution. Fig. 9 plots several quantities, with brackets representing expectations over a batch of trajectories and $E_0(\mathbf{x}) = 1/2x^2$. First, we see that the Hamiltonian is approximately conserved. The average energy starts high then quickly converges to a low energy state, while the kinetic energy $(d\, r(t))$ starts low and becomes large. Finally, we see that the weights in Eq. 8 correctly give an estimate of the partition function. Note that at $t = 0$, the weights correspond to standard importance sampling.

## C.3 Training Energy-Based Models

Because the Jarzynski sampler is weighted, we cannot estimate the MMD metrics used in Sec. 3 in the same way. Instead, we test the approach by training an energy model to match a distribution with

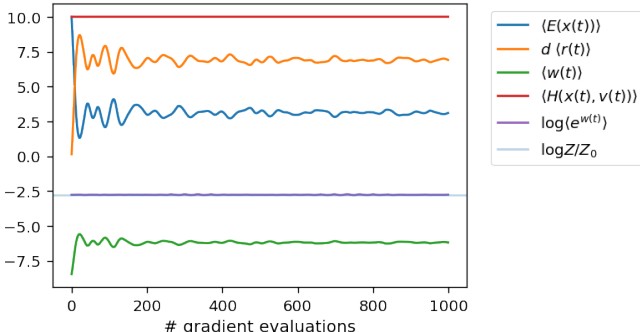

Figure 9: Plotting various quantities for ESH dynamics with a random Gaussian as the energy model.

a known ground truth. We consider an energy model, $p(\mathbf{x}) = e^{-E_\theta(\mathbf{x})}/Z$, where $E_\theta(\mathbf{x})$ is specified by a neural network. Estimating gradients to train such a model to maximize the likelihood of the data requires sampling from the energy model (App. B.3). Fig. 8 visualizes the energy landscape learned from data sampled from a 2-D distribution. The architecture and hyper-parameters are taken from [21] and included in App. D.3. Samplers are initialized with noise and then run for 50 steps per training iteration, using the Jarzynski sampler for ESH. Better sampling leads to better gradients and therefore the model trained with ESH sampling more crisply matches the true distribution.

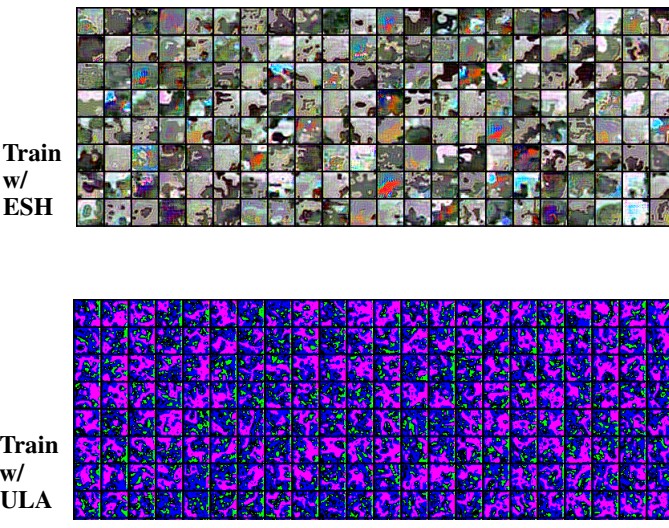

**Train w/ ESH**

**Train w/ ULA**

Figure 10: Results training neural energy models using ULA versus ESH sampling with 500k gradient evaluations. (Top) Learned energy for synthetic datasets with noise initialization. (Bottom) Examples from the training buffer using persistent contrastive divergence on CIFAR-10.

**Unstable Regime for Langevin Dynamics**    Training EBMs with Langevin dynamics for sampling is difficult and requires a great deal of hyper-parameter search to succeed [49; 48; 21; 22; 13]. For the experiment in Fig. 10, we wanted to demonstrate that the unbiased Jarzynski sampler based on ESH dynamics could lead to stable training in a regime where training with Langevin dynamics is unstable. To that end, we considered a large model for CIFAR, with short chains (50 steps), and we omitted heuristics like adding data noise that are typically used to improve stability. We initialized chains using persistent contrastive divergence with a large buffer. All hyper-parameters are listed in Sec. D.4. Predictably, the EBM trained with Langevin dynamics fails egregiously without carefully

chosen hyper-parameters. The training with ESH dynamics was stable and led to visually plausible color distributions and diverse signals. However, although training was stable, we noticed that the quality of the solutions stopped improving after the early epochs. We now discuss the reason.

**Bias-Variance Trade-off for Jarzynski Sampling**    Methods like importance sampling use weights to re-weight samples to make one distribution resemble another. A core issue with this approach is that if some weights are much larger than others, then samples that are rare but have very high weight may be missed in a given batch of samples. This increases the variance of the estimator. Although the Jarzynski ESH sampler is an unbiased estimator for expectations under the energy model, it may have very high variance if the weights are very unbalanced. For the 2-d examples in Fig. 8 this was not an issue. However, when training high-dimensional energy models for CIFAR, it was an issue. Fig. 11 shows the maximum weight per batch for training an EBM for CIFAR. After a few iterations, almost all the weight ends up concentrated on a single image in each batch. This leads to a high variance estimator with noisy gradient estimates. In contrast, using persistent contrastive divergence will have higher bias, but because all samples are equally weighted it will also have lower variance.

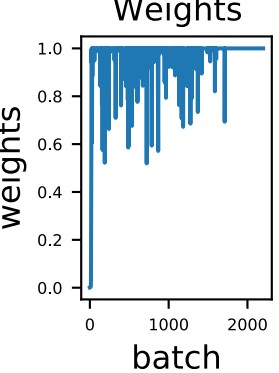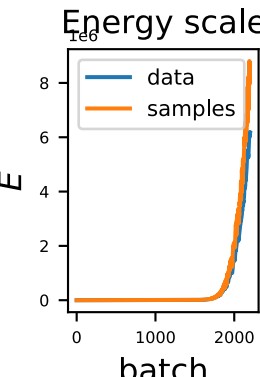

Figure 11: The maximum weight per batch over time for Jarzynski ESH sampling during CIFAR-10 energy-based model training. We also show the average energy for batches of data versus sampled batches.

# D    Algorithms and Implementation

## D.1    Leapfrog Integrator for ESH Dynamics

The algorithm for the leapfrog integrator of ESH dynamics in the time-scaled coordinates is given in Alg. 1. For implementation, we can improve numerical accuracy by scaling the numerator and denominator of the $\mathbf{u}$ updates in Eq. 7 by $e^{-2|\mathbf{g}|\epsilon/d}$, so that we are more likely to get underflows than overflows in the exponentials. We also found that we had to treat $\mathbf{u} \cdot \mathbf{e} = -1$ as a special case, where the update simplifies to $\mathbf{u}(t + \epsilon) = -\mathbf{e}$. A PyTorch implementation is available at https://github.com/gregversteeg/esh_dynamics.

We simulate in time-scaled coordinates on a regular grid, $\bar{t} = 0, \epsilon, 2\epsilon, \ldots, \bar{T}$. We can easily convert from scaled to un-scaled time using the relation $dt = d\bar{t}\,|\mathbf{v}(\bar{t})|/d$ which gives us the following.

$$t(\bar{t}) = \int_0^{\bar{t}} d\bar{t}'|\mathbf{v}(\bar{t}')|/d$$

We can also convert solutions in terms of $\mathbf{v}$ or $\mathbf{u}, r$ using Eq. 6.

**Algorithm 1** Time scaled leapfrog integrator for ESH dynamics.

---

**Require:** $E(\mathbf{x})$          # Target energy to sample
**Require:** $\epsilon, N$          # step size, number of steps
 1: $\bar{T} = \epsilon N$          # total time
 2: Initialize $\mathbf{x}(0) \sim \mathcal{N}(0, I)$ or accept as input
 3: $\mathbf{u}(0) \sim \mathcal{N}(0, I)$, $r(0) = 0$
 4: $\mathbf{u}(0) = \mathbf{u}(0)/|\mathbf{u}(0)|$
 5: $\mathbf{g}(\mathbf{x}) = \partial_{\mathbf{x}} E(\mathbf{x})$          # Gradient of energy
 6: **for** $i = 1 \ldots N$ **do**
 7:      # Eq. 15 updates
 8:      $\mathbf{u}(\bar{t} + \epsilon/2) = \mathbf{f}(\mathbf{x}(\bar{t}), \mathbf{u}(\bar{t}), \epsilon/2)$      # Half step $\mathbf{u}$
 9:      $r(\bar{t} + \epsilon/2) = r(\bar{t}) + a(\epsilon/2, \mathbf{g}(\mathbf{x}(\bar{t})))$      # Half step $r$
10:      $\mathbf{x}(\bar{t} + \epsilon) = \mathbf{x}(\bar{t}) + \epsilon \mathbf{u}(\bar{t} + \epsilon/2)$      # Full step in $\mathbf{x}$
11:      $\mathbf{u}(\bar{t} + \epsilon) = \mathbf{f}(\mathbf{x}(\bar{t} + \epsilon), \mathbf{u}(\bar{t} + \epsilon/2), \epsilon/2)$      # Half step $\mathbf{u}$
12:      $r(\bar{t} + \epsilon) = r(\bar{t}) + a(\epsilon/2, \mathbf{g}(\mathbf{x}(\bar{t} + \epsilon)))$      # Half step $r$
13: **end for**
14: **return** $\mathbf{x}(0), \ldots, \mathbf{x}(\bar{T}), \mathbf{u}(0), \ldots, \mathbf{u}(\bar{T}), r(0), \ldots, r(\bar{T})$

---

### D.2 Sampling Algorithms Using ESH Dynamics

Recalling from Sec. 2.2, we relate expectations under our target distribution to expectations over trajectories using ergodicity.

$$\mathbb{E}_{\mathbf{x} \sim e^{-E(\mathbf{x})}/Z}[h(\mathbf{x})] \overset{\text{Eq. 3}}{=} \mathbb{E}_{\mathbf{x} \sim p(\mathbf{x}, \mathbf{v})}[h(\mathbf{x})] \overset{\text{Eq. 4}}{\approx} \mathbb{E}_{t \sim \mathcal{U}[0,T]}[h(\mathbf{x}(t))]$$

The expression on the right leads to a method for ergodic sampling in Alg. 2.

However, in general we may not want to store the entire trajectory of samples, especially in high-dimensional spaces like images. This prompts us to construct a slightly different estimator.

$$\mathbb{E}_{t \sim \mathcal{U}[0,T]}[h(\mathbf{x}(t))] = \frac{1}{T} \int_0^T dt \, h(\mathbf{x}(t))$$

$$\mathbb{E}_{t \sim \mathcal{U}[0,T]}[h(\mathbf{x}(t))] = \frac{1}{\bar{T}} \frac{\bar{T}}{T} \int_0^{\bar{T}} d\bar{t} \, |\mathbf{v}(\bar{t})|/d \, h(\mathbf{x}(\bar{t}))$$

$$= \mathbb{E}_{\bar{t} \sim \mathcal{U}[0,\bar{T}]}[\mathbf{v}(\bar{t})/D \, h(\mathbf{x}(\bar{t}))]$$

$$\text{where} \quad D \equiv \frac{1}{\bar{T}} \int_0^{\bar{T}} d\bar{t} \, |\mathbf{v}(\bar{t})|/d = \mathbb{E}_{\bar{t} \sim \mathcal{U}[0,\bar{T}]}[\mathbf{v}(\bar{t})]$$

If we discretize this estimator, we get the following.

$$\mathbb{E}_{t \sim \mathcal{U}[0,T]}[h(\mathbf{x}(t))] \approx \sum_{i=0}^N w_i \, h(\mathbf{x}(\bar{t}_i)) \quad \text{with} \quad w_i \equiv \frac{|\mathbf{v}(\bar{t}_i)|}{\sum_{j=0}^N |\mathbf{v}(\bar{t}_j)|}$$

Now we can draw samples according to this normalized weight to sample from the target distribution. Instead of storing the entire sequence of weights and states, $w_i, \mathbf{x}(\bar{t}_i)$, we can sample by storing a single sample/image in a buffer and updating in an online way by replacing it with the next sample with some probability. This classic technique is called reservoir sampling [34]. The final state of the buffer will correspond to a state drawn according to the weights, $w$. The procedure is shown in Alg. 3.

| **Algorithm 2** Ergodic sampling | **Algorithm 3** Reservoir sampling [34] |
|---|---|
| **Require:** $\mathbf{x}(\bar{t}_1), \ldots, \mathbf{x}(\bar{t}_N)$ from Alg. 1 | **Require:** $\epsilon, N, E$  # step size, steps, Energy |
| **Require:** $r(\bar{t}_1), \ldots, r(\bar{t}_N)$ from Alg. 1 | 1: $\mathbf{x}^* = \emptyset$   # (Empty) buffer for current sample |
| 1: **for** $i = 1 \ldots N$ **do** | 2: Cum-weight = 0 |
| 2:   $t_i = \int_0^{\bar{t}_i} d\bar{t}' e^{r(\bar{t}')}/d$ | 3: **for** $i = 1 \ldots N$ **do** |
| 3: **end for** | 4:   $t, \mathbf{x}, \mathbf{v} = $ Alg 1$(E, \epsilon, 1)$ # 1 step of dynamics |
| 4: $T = t_N$ | 5:   Cum-weight = Cum-weight + $|\mathbf{v}|$ |
| 5: Select $t^*$ randomly from $[0, T]$ | 6:   $\mathbf{x}^* \leftarrow \mathbf{x}$ with probability $|\mathbf{v}|/$Cum-weight |
| 6: Linearly interpolate to get $\mathbf{x}^* \equiv \mathbf{x}(t^*)$ | 7: **end for** |
| 7: **return** $\mathbf{x}^*$   # $\mathbf{x}^* \sim p(\mathbf{x})$ if ergodicity holds | 8: **return** $\mathbf{x}^*$   # sample from $p(\mathbf{x}) = e^{-E(\mathbf{x})}/Z$ |

Jarzynski sampling has a different interpretation based on normalizing flows and importance sampling, as described in Sec. C. The algorithm is summarized in Alg. 4.

| **Algorithm 4** Jarzynski sampling |
|---|
| **Require:** $E_0, E$  # Initial energy and target energy |
| 1: $(\mathbf{x}^1(0), \ldots, \mathbf{x}^n(0)) \sim e^{-E_0(\mathbf{x}(0))}$   # Batch size $n$ |
| 2: $\mathbf{x}^j(t), \mathbf{u}^j(t), r^j(t)$ from Alg. 1 |
| 3: $w^j(t) = E_0(\mathbf{x}^j(0)) - E(\mathbf{x}^j(0)) + r^j(t)$   # Eq. 8 |
| 4: $\bar{w}^j(t) = e^{w^j(t)}/\sum_i e^{w^i(t)}$    # Self normalized importance sampling |
| 5: **return** $\mathbf{x}^j(t), \bar{w}^j(t)$   # Weighted samples are unbiased estimate of samples from $p(\mathbf{x}) = e^{-E(\mathbf{x})}/Z$ |

### D.3   Model Architecture

We use the following architectures from [21] for training EBMs. Convolutional operation $conv(n)$ with $n$ output feature maps and bias term and $fc(n)$ is a fully connected layer. Leaky-ReLU nonlinearity $LReLU$ with default leaky factor 0.05. We set $n_f = 128$.

| Energy-based Model ($32 \times 32 \times 3$) | | |
|---|---|---|
| Layers | In-Out Size | Stride |
| Input | $32 \times 32 \times 3$ | |
| $3 \times 3$ conv($n_f$), LReLU | $32 \times 32 \times n_f$ | 1 |
| $4 \times 4$ conv($2 * n_f$), LReLU | $16 \times 16 \times (2 * n_f)$ | 2 |
| $4 \times 4$ conv($4 * n_f$), LReLU | $8 \times 8 \times (4 * n_f)$ | 2 |
| $4 \times 4$ conv($8 * n_f$), LReLU | $4 \times 4 \times (8 * n_f)$ | 2 |
| $4 \times 4$ conv(1) | $1 \times 1 \times 1$ | 1 |

Table 2: Network structures ($32 \times 32 \times 3$).

| Energy-based Model $(2, )$ | |
|:---:|:---:|
| Layers | In-Out Size |
| Input | 2 |
| fc($n_f$), LReLU | $n_f$ |
| fc($2 * n_f$), LReLU | $(2 * n_f)$ |
| fc($2 * n_f$), LReLU | $(2 * n_f)$ |
| fc($2 * n_f$), LReLU | $(2 * n_f)$ |
| fc(1) | 1 |

Table 3: Network structure for 2-D toy datasets.

## D.4 Hyper-parameters

**Table 3.2 and Fig. 15 Experiments** For HMC, we used a step size of $\epsilon = 0.01$ with $k = 5$ steps per chain. For MALA and ULA we used both step sizes of $\epsilon = \{0.01, 0.1\}$. For NUTS we used the default settings from the littleMCMC implementation linked in Sec. D.6. For the Nose-Hoover sampler we used a step size of $\epsilon = 0.01$ because larger step sizes resulted in numerical errors. For the ESH Leapfrog sampler we used $\epsilon = 0.1$ because we noted from the comparison of ESH ODE solvers that smaller values were less effective, and $\epsilon = 1$ sometimes led to numerical errors.

**Sampling pre-trained JEM model** We used the same step size used for training the JEM model for both MALA and ULA, $\epsilon = 0.01$. The Langevin dynamics is defined as,

$$\mathbf{x}_{t+1} = \mathbf{x}_t + \epsilon^2/2\mathbf{g}(\mathbf{x}_t) + \epsilon \mathbf{v}_t,$$

where $\mathbf{v}_t \sim \mathcal{N}(0, 1)$ at each step. However, in JEM and EBM training the energy model and its gradients are implicitly scaled according to the step size, $E(\mathbf{x}) \to E(\mathbf{x})/(\epsilon^2/2)$ so that the effective update is as follows.

$$\mathbf{x}_{t+1} = \mathbf{x}_t + \mathbf{g}(\mathbf{x}_t) + \epsilon \mathbf{v}_t.$$

For sampling with other methods, we also scaled the energy function as is done for Langevin dynamics. For the ESH leapfrog sampler we used step sizes of $\epsilon = \{0.5, 1\}$.

**Training EBMs on 2-D Toy Models** We use the architecture in Table 3 with $n_f = 32$ for all experiments training neural energy models on 2-D datasets. We trained for 10000 iterations with a batch size of 100. We used SGD for optimization with a learning rate of 0.1. We ran each chain for 50 steps. For rings, we initialized from a scaled unit Gaussian with variance 4 and for the mixture of Gaussians we initialized from the uniform distribution in $[-2, 2]$. As suggested by [22], we used ULA for sampling with a step size of $\epsilon = 0.1$. For ESH we also used a step size $\epsilon = 0.1$.

**CIFAR EBM details** For the experiment training an EBM on CIFAR data in Sec. 3.4 we chose to match hyper-parameters as closely as possible to EBM training with Langevin dynamics in a regime where it is known to be stable. We use the architecture in Table 2 with $n_f = 128$. We add Gaussian noise with variance 0.03 as done by [21] to stabilize training. We use chains with 100 steps each. We used a replay buffer of size $10,000$. Following [49] we applied spectral normalization [54], and we also ensemble by averaging the energy over the last 10 epoch checkpoints. The optimizer was ADAM with a learning rate of 0.0002 with $\beta_1 = 0.5, \beta_2 = 0.999$. The buffer was initialized uniformly over $[-1, 1]$. We used the same scaled energy function for both Langevin and ESH sampler. The batch size was 500 and we trained for 250 epochs. ULA step size was $\epsilon = 0.01$ and ESH step size was $\epsilon = 0.5$.

**Test time sampling** In Fig. 7 we show samples generated in exactly the same manner as training, except with a fixed energy function. For sampling from scratch, we initialize the sampler with an equal mixture of a random constant color and per pixel standard normal noise. The random constant color is drawn from a normal distribution constructed from the pixel-averaged image color of CIFAR training images. We also clipped color values to be in the range $[-1, 1]$.

**Jarzynski Sampler EBM Training in Sec. C.3** We use the architecture in Table 2 with $n_f = 128$ for training a neural energy model on CIFAR data in Sec. 3.4. We did not add noise to the data as was done by [21] to stabilize training, to highlight that this step is not necessary for unbiased Jarzynski samplers. We use short chains with 50 steps each. We used a replay buffer with the same size as the training data, $50,000$. We trained for 30 epochs over the training data with batch sizes of 1000. The optimizer was ADAM with a learning rate of $10^{-3}$ and $\beta_1 = 0.9, \beta_2 = 0.999$. Chains were initialized from a persistent buffer that was initialized uniformly over $[-1, 1]$. Note that when we initialize ESH dynamics with a sample from the buffer an ambiguity arises in setting the value of $E_0(\mathbf{x}(0))$ in Eq. 8. We considered $E_0$ to be constant. For the Langevin sampler we scaled the energy function by $2/\epsilon^2$, but for ESH sampling we skipped this step to show it is not necessary. ULA sampling used a step size of $\epsilon = 0.01$ (with larger gradient steps due to energy scaling), and ESH used a step size $\epsilon = 0.5$. Although a step size for ESH of $\epsilon = 1$ worked better for sampling the pre-trained JEM model, the error from discretizing the ODE goes like $O(\epsilon^3)$ so we erred on the conservative side by using $\epsilon < 1$. When we tried $\epsilon > 1$ we always got numerical errors, so the apparent stability of the choice $\epsilon = 1$ may require more consideration.

### D.5 Computational Cost

Our small experiments comparing samplers and ablation study were done on a single NVIDIA 1080Ti GPU and took from minutes to hours. The Jarzynski sampler trained CIFAR models were trained using 4 Tesla V100 GPUs for about 3 hours for each experiment. The CIFAR EBM in Sec. 3.4 trained using 4 Tesla V100 GPUs for about 1 day for each experiment.

### D.6 Code Used and Licenses

Our code is provided at `https://github.com/gregversteeg/esh_dynamics`. We implemented our approach in PyTorch and also used code from a number of sources.

| Link | Citation | License |
|------|----------|---------|
| `https://github.com/point0bar1/ebm-anatomy` | [21] | MIT License |
| `https://github.com/rtqichen/torchdiffeq` | [44] | MIT License |
| `https://github.com/wgrathwohl/JEM` | [48] | Apache version 2 |
| `https://github.com/eigenfoo/littlemcmc` | [4] | Apache version 2 |

## E  Additional Results

### E.1 Comparing Integrators for ESH

We provide additional figures about how different ESH dynamics integrators perform. In Fig. 12 we show how the adaptive step size in the Runge-Kutta integrator correlates with velocity, motivating our attempt to scale the ESH dynamics. Fig. 13 shows the evolution of a batch of 500 chains evolving using different integrators. You can see that the ESH leapfrog with scaled dynamics explores the space much more rapidly than the other methods. In Fig. 15 we show the MMD scores for various integrators on our test sets. The scaled leapfrog integrator dominates across all experiments.

In Fig.16 we show the total Hamiltonian and kinetic energy for different ESH integrators. They all approximately conserve the Hamiltonian, but only the leapfrog integrator approaches a low energy, high kinetic energy solution quickly. Quickly approaching this regime followed by small oscillations in kinetic energy is a typical characteristic of the dynamics.

### E.2 Visualizing Ergodicity

To empirically visualize whether the ESH dynamics are ergodic, we simulated a single long trajectory on one of our 2D datasets, MOG. We used a small step size, $\epsilon = 0.001$, to accurately simulate the dynamics. It is possible that larger step sizes introduce errors in the trajectory, but that these errors actually help the dynamics mix. The reason is that the dynamics are supposed to sample the target distribution for any energy from any initial point. Therefore, discretization error may actually help by mixing between different trajectories, any of which ergodically sample the target distribution. We want to be sure that the mixing comes from the dynamics, and not discretization error, for this test. Then, to quantify how well the dynamics converge to the correct distribution, we

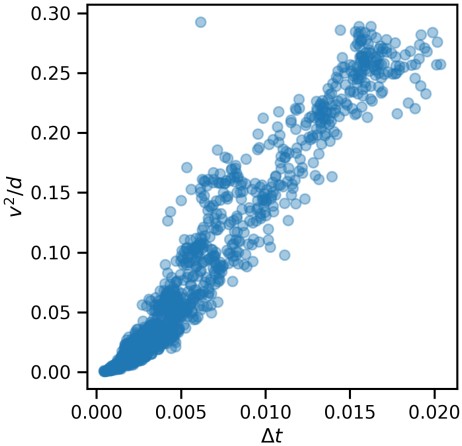

Figure 12: In the original unscaled ODE, Eq. 2, the adaptive time-step from the Runge-Kutta integrator is correlated with the magnitude of the velocity. This motivates our idea to re-scale the time-step.

ergodically sample 500 points and estimate the MMD to samples from the ground truth distribution as in Sec. 3. As shown in Fig. 17, the trajectory seems to be ergodic, i.e. it visits all points in the space and does not get stuck in a subspace. This is quantified by the MMD which is close to zero as expected. The trajectory does not appear to have any regularity or cycles. Note that ergodicity does not necessarily imply chaotic dynamics, which require that small changes in initial conditions lead to strongly diverging trajectories as measured by Lyapunov exponents, for instance. We did not test how chaotic the ESH dynamics are, but this would be interesting to study in future work, especially as it relates to mixing speed for sampling.

### E.3 Funnel

All methods failed on the Funnel dataset. The MMD plots in Fig. 18 are ambiguous, but from looking at the evolution we see that none of the samplers have captured the funnel shape. For a dataset where the length scales differ by orders of magnitude, auxiliary models or second order methods may be justified.

### E.4 Sampling a Pre-trained JEM Model with Informative Prior

In Fig. 6, we showed that ESH samplers found much lower energy solutions than other samplers when initialized from noise with a pre-trained neural network energy model. However, JEM was trained with persistent contrastive divergence, so it makes sense to compare the solutions found when starting from a sample in the persistent buffer. Fig. 19 shows that the result is the same, ESH finds much lower energy solutions.

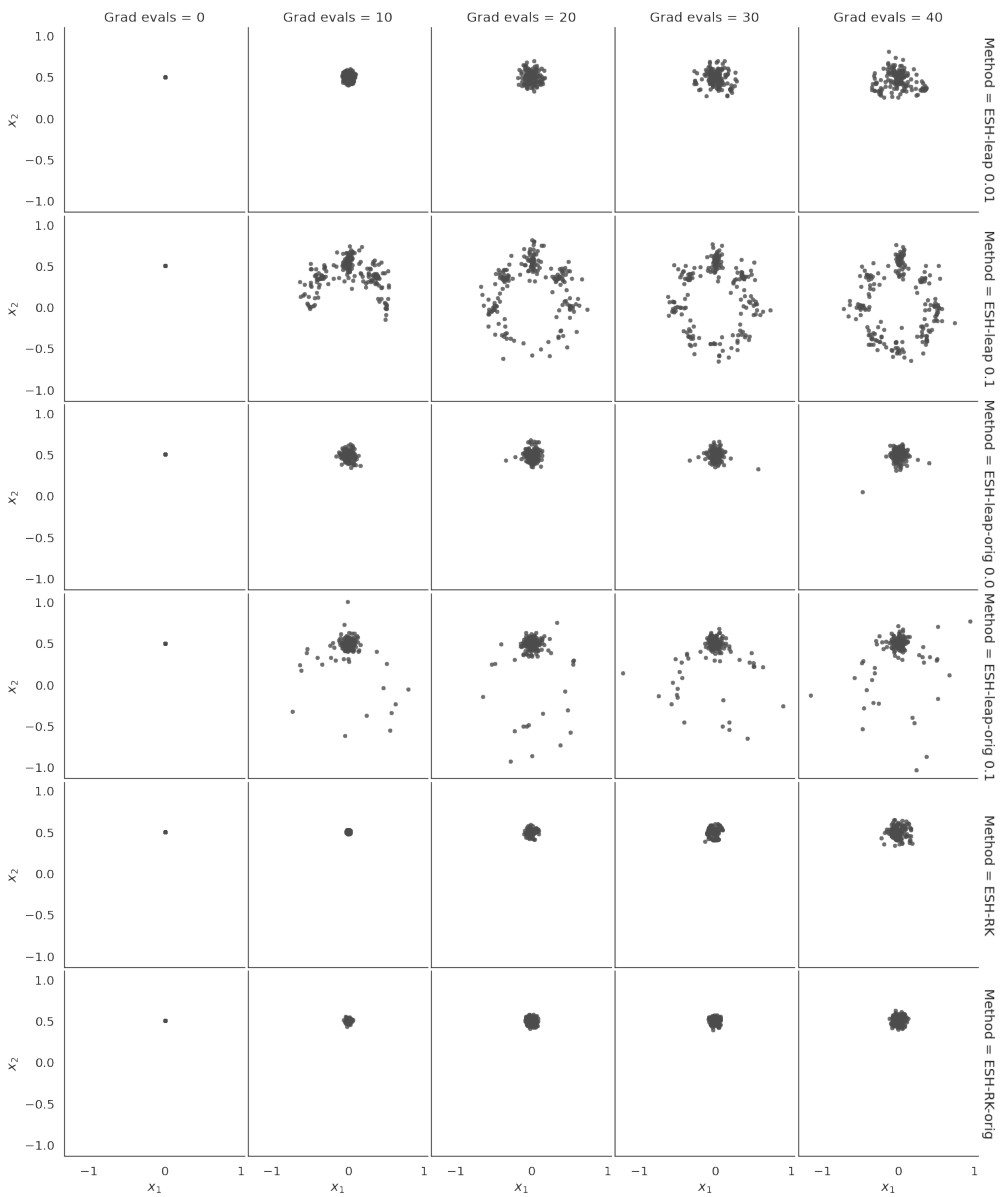

Figure 13: For the 2d mixture of Gaussians with an informative prior, we visualize the distribution of 500 chains after various numbers of gradient evaluations, comparing different ESH dynamic integrators.

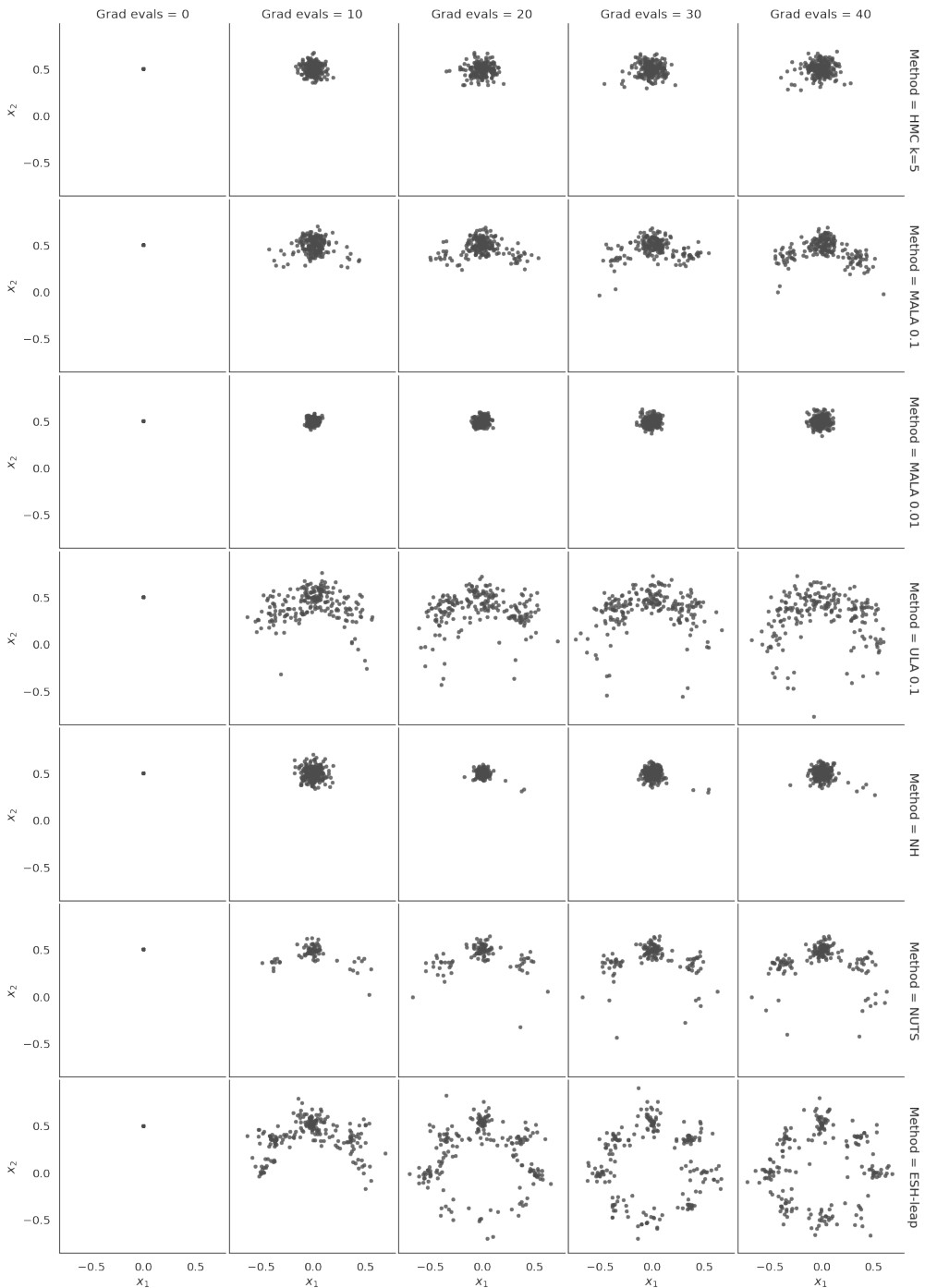

Figure 14: For the 2d mixture of Gaussians with an informative prior, we visualize the distribution of 500 chains after various numbers of gradient evaluations, comparing different samplers.

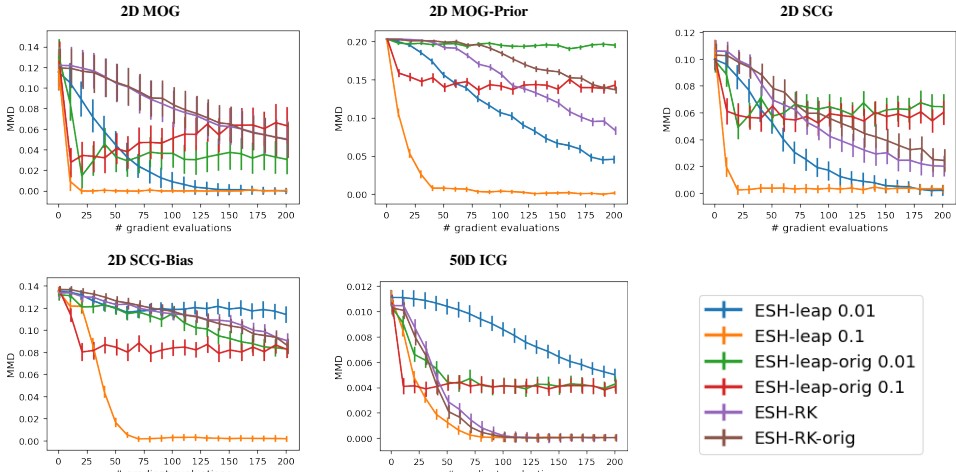

Figure 15: Maximum Mean Discrepancy (MMD) as a function of the number of gradient evaluations for other datasets using different ESH integrators.

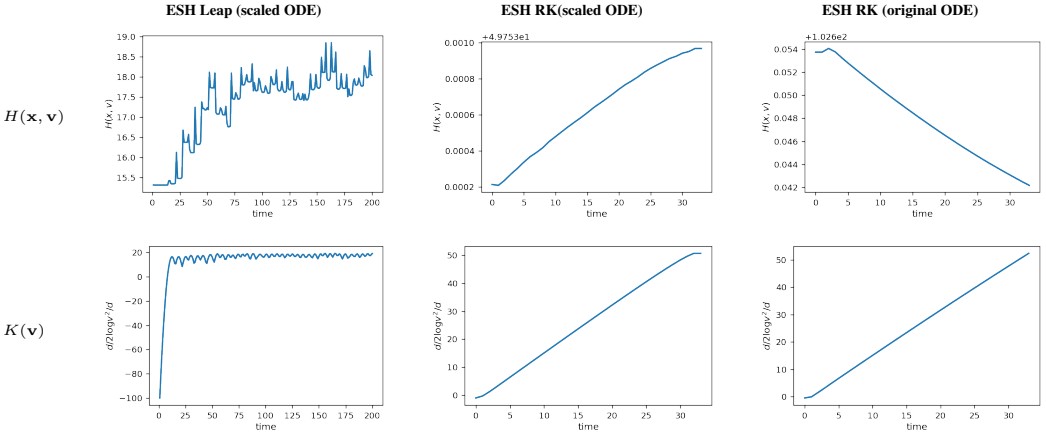

Figure 16: For the MOG dataset, we visualize the total Hamiltonian and the kinetic energy for different ESH integrators. Note the small relative scale on the $y$-axis for the total Hamiltonian. The step size is $\epsilon = 0.1$.

MMD = 0.00541 , 95% CI = [0.00827,0.01350]

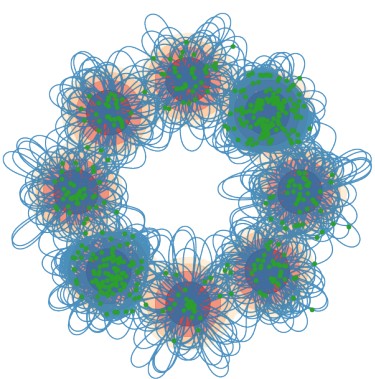

Figure 17: For the 2D MOG dataset, we use ESH-Leap to simulate a long trajectory $(500, 000$ steps) with small step size $(\epsilon = 0.001)$ for a precise simulation. We ergodically sample 500 points from this single trajectory and compute the MMD to the ground truth.

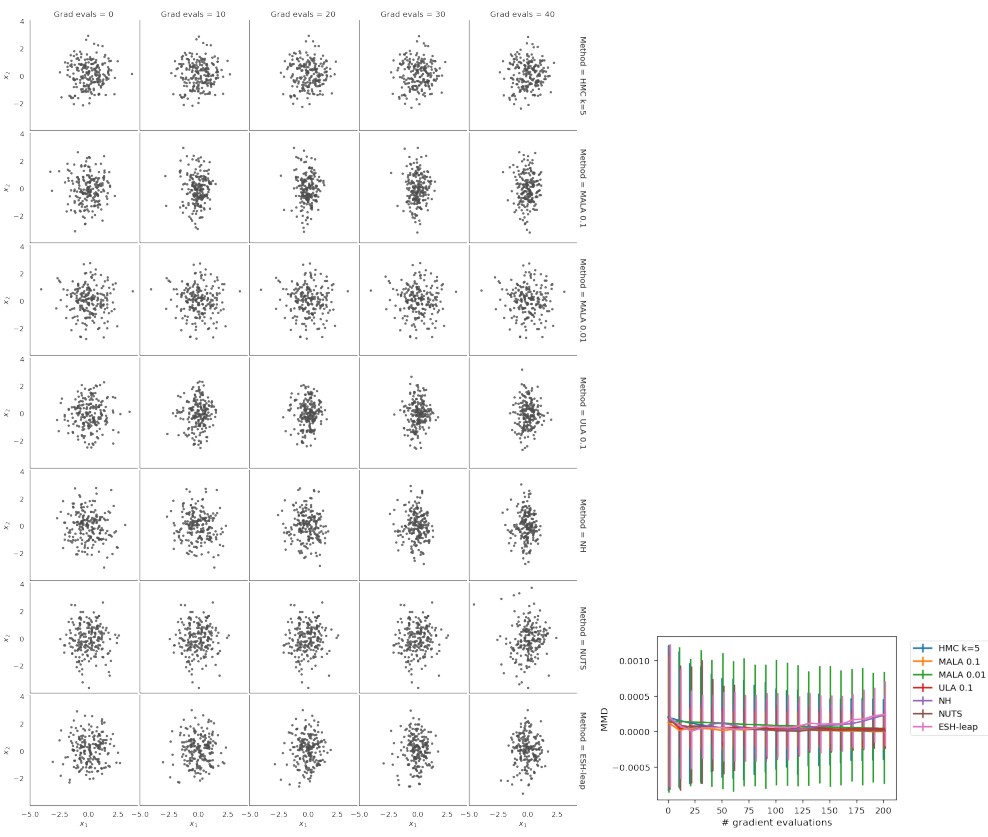

Figure 18: For the funnel dataset, $x_1$ ($x$-axis) controls the width of the funnel. We should see a thin funnel toward the left of the plot. We visualize the distribution of 500 chains after various numbers of gradient evaluations.

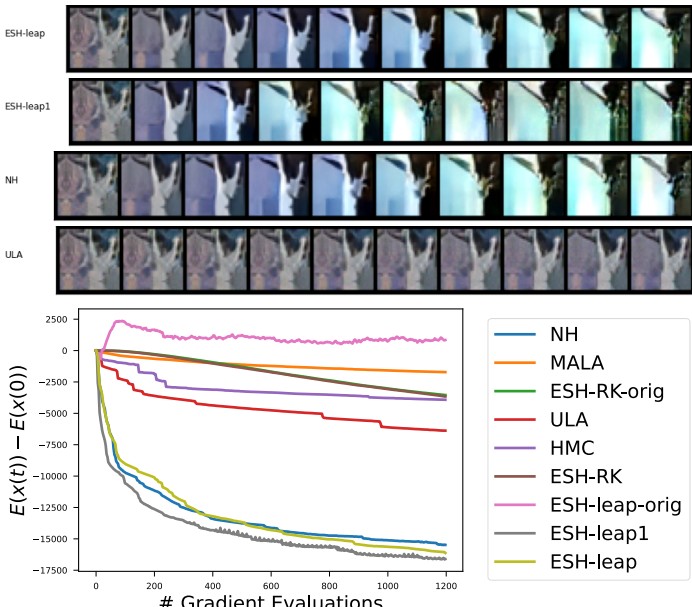

Figure 19: (Top) Example of sampling chains from replay buffer initialization with 200 gradient evaluations per method. (Bottom) Average energy over time for a batch of 50 samples using different samplers.