# OpenReview forum: "Hamiltonian Dynamics with Non-Newtonian Momentum for Rapid Sampling"
_NeurIPS.cc/2021/Conference — NeurIPS 2021 Poster_

### Official Review · Reviewer_T1qm · 2021-07-13

**Rating:** 7
**Confidence:** 4

**Summary:**

This paper proposes a deterministic sampling dynamic, rapid energy sampling Hamiltonian dynamics (ESH). Since there is no stochasticity, there is no rejection. The kinetic energy is formulated in a way that the dynamic stays at a state for the time proportional to the Boltzmann distribution of the (potential) energy. This paper also propose a leapfrog integrator specialized for ESH. Compared to MCMC approaches, ESH opens a way of much faster sampling from unnormalized energy based models without any auxiliary generating models.


**Limitations And Societal Impact:**

Yes

**Main Review:**

This submission looks almost complete. I think the idea of defining a non-Newtonian momentum is very novel. This paper may bring about a paradigm shift in how the machine learning community approaches the problem of sampling from unnormalized models. Their empirical performance in mixing between modes (Fig 1, Fig 3) is very impressive. They only have one hyperparameter to tune, epsilon, which is very simple compared to other MCMC approaches although their algorithm itself is far from being simple: Langevin is hard to tune, but implementation is very simple with only a few lines. But the submitted code for time adapted leap frog sampler was daunting.


Questions:

First, what happens to your very intuitive explanation in Fig. 2 and line 72-81, when integrating the scaled time dynamics? Your K is set to make the time spend around x as 1/\dot{x} (line 2 in Figure 2). But, now again the time is scaled  by 1/|v|.. It is kind of unclear what’s going on and what is the impact of it when we re-scale back.

For line 86, what happens if the support of p(x) is unbounded, e.g. R^d? We cannot define a uniform distribution here, and in your formulation the identity in (3) seems crucial. Even then would your method be established?

In (7), how do I obtain u(t+epsilon/2) in the third line? Is it just by applying epsilon/2 in place of the epsilon in the first line?

In 120-121, why the effective step size can vary dramatically depending on the magnitude of the velocity? Could you give me some intuitive explanation on it?

In 254, a correct label may not be Fig 13 (funnel) but Fig 17 or another? Section 3.4. did not even specify which data was used. Therefore, it adds confusion when you say “as in a first experiment” (line 666). What is the relationship between the experiments in section 3.4 and D.3. Are they separate, different experiments? It is confusing because in D.3, you have “experiment 1” and “experiment 2”, making unclear if they are additional details of the experiment in 3.4 or else separate, additional experiments.

In 272, is there any (intuitive) reason/explanation why you initialized the sampler by using Jarzynski, not by other two algorithms or leap-orig?

Which algorithm is used for the experiment? I thought you are using Algorithm 1 or a version without time adaptation, and sometimes Jarzynski for initialization (line 272). But, when addressing bias-variance trade-off, in 676-688, Jarzynski ESH sampler is mainly discussed. Then, added in 687 that you use this ignoring-the-Jarzynski-weight technique. Please identify clearly which algorithm is used in each of your experiments…


In Fig 15, how did your replay buffer images look like? If the buffer was already reasonably realistic, what is the implication of this figure?

In file samplers.py, you did not recommend to use your submitted code for anything because it works with a single chain, not parallel chains. But, for the computational cost, in line 640, only minutes or hours were taken with only a single GPU. Did the cost calculation come from your single chain code?

Is there any reason that you did not show the full trained CIFAR model by using ESH? You stopped learning at epoch 100 and only compared at that moment.



**Time Spent Reviewing:**

4

---

> ### Author Response · Authors · 2021-08-06
> **Code review and reference catch thanks**
>
> First, we want to give an extra thanks for giving feedback on the code, it is very helpful and goes above and beyond what is required for reviewers. We’ll respond to specific points below.
>
> > First, what happens to your very intuitive explanation in Fig. 2 and line 72-81, when integrating the scaled time dynamics? Your K is set to make the time spend around x as 1/\dot{x} (line 2 in Figure 2). But, now again the time is scaled by 1/|v|.. It is kind of unclear what’s going on and what is the impact of it when we re-scale back.
>
> Yes, it is a bit confusingly written. In the unscaled dynamics the “velocity” gets large, but the updates in position (x) slow down. This is numerically very bad because it takes many updates to make progress in x. Therefore, we scale the dynamics so that the updates in x always have the same length. But then to take ergodic averages over (unscaled) time in a uniform way (eq. 4), we have to put in a term to correct for the scaled dynamics (a change of variables *hidden* in line 130). A more explicit derivation of this term is buried in Algorithm 2 in the appendix. If we have more space in the final version, we would like to move and expand some of that discussion in the main text.
>
> > For line 86, what happens if the support of p(x) is unbounded, e.g. R^d? We cannot define a uniform distribution here, and in your formulation the identity in (3) seems crucial. Even then would your method be established?
>
> The domain of x can be unbounded, and p(x) is still normalizable (e.g. a Gaussian). You probably meant the uniform distribution over the delta function for p(x,v) in Eq. 3. Formally, it seems that this would have to be normalizable to be well-defined, but the final result does not depend on $Z'$ so it's not clear if it would be a deal-breaker (i.e., we can imagine a limit where Z' goes to infinity, but since the final result doesn't depend on Z' it may not matter). We will consider this more, but wanted to respond to the other points right away so that there is opportunity for discussion.
>
> > In (7), how do I obtain u(t+epsilon/2) in the third line? Is it just by applying epsilon/2 in place of the epsilon in the first line?
>
> Yes, that’s correct, we had to do it that way to save space. It is more clear to write the leapfrog steps explicitly.
>
> > In 120-121, why the effective step size can vary dramatically depending on the magnitude of the velocity? Could you give me some intuitive explanation on it?
>
> Yes, because the steps in x are proportional to 1/v^2 in Eq. 5, if the velocity is near to zero x can move very large distances. This happens often as a particle moves out of one low energy mode and is poised to tip into another, the magnitude of the velocity is small. Conversely, the particle picks up speed as it enters a low energy mode, and then the updates in Eq. 5 are small.
>
> > In 254, a correct label may not be Fig 13 (funnel) but Fig 17 or another?
>
> Great catch, thank you, that should have been Fig. 17.
>
> > Section 3.4. did not even specify which data was used. Therefore, it adds confusion when you say “as in a first experiment” (line 666). What is the relationship between the experiments in section 3.4 and D.3. Are they separate, different experiments? It is confusing because in D.3, you have “experiment 1” and “experiment 2”, making unclear if they are additional details of the experiment in 3.4 or else separate, additional experiments.
>
>  We will clarify the experiments on CIFAR-10. The one in Sec. 3.4 is “Experiment 1” (with details described in the appendix D.3) and the additional figures in D.3 show results from experiment 2. We will expand on this and make it clear in the main text.
>
> > In 272, is there any (intuitive) reason/explanation why you initialized the sampler by using Jarzynski, not by other two algorithms or leap-orig?
>
> The motivation was that for images, it is expensive to store the entire trajectory of images, and then to sample ergodically from them. In the Jarzynski sampler, you always store and use the last image in the trajectory, with some weighting factor. However, we didn’t realize until after submission that there is a simple reservoir sampling scheme that would let us use the leapfrog ergodic sampler directly without storing a trajectory of images and with minimal modifications.
>
> > Which algorithm is used for the experiment?
>
> We will clarify this. For images, we always used Jarzynski-type estimators for the reason described in the last answer. However, for experiment 2, we found that an unweighted version of the Jarzynski estimator worked better because it had lower variance. This is related to our response to the last point raised by Reviewer daEc.
>
> > In Fig 15, how did your replay buffer images look like? If the buffer was already reasonably realistic, what is the implication of this figure?
>
> If you zoom in, the buffer images look very good for both ULA and ESH. The surprising thing was that if you sample from random initialization you get very different looking images for ULA and ESH (Fig. 16). This is in line with observations from Nijkamp et al where EBMs trained with PCD can give good images in the buffer, but “long-run” samples from the energy model initialized from scratch are actually quite bad. This signals poor convergence of the energy model.
>
> > In file samplers.py, you did not recommend to use your submitted code for anything because it works with a single chain, not parallel chains. But, for the computational cost, in line 640, only minutes or hours were taken with only a single GPU. Did the cost calculation come from your single chain code?
>
> Samplers.py implements the code for a number of different comparisons used in Sec. 3.1 and 3.2. Those experiments are smaller scale and the entire script takes about 30 minutes on a CPU, even processing single chains at a time (because we had a hard time adapting the ODE solver to multiple chains). We will add the precise number to the computational cost section.  The most computationally intensive experiments, by far, was training the neural network energy-based models.
> The ESH code that processes many chains in parallel, on CPU or GPU, is provided in esh_leap.py. The “esh_leap_step” is relatively simple, and then the other methods wrap that in different ways.
>
> > Is there any reason that you did not show the full trained CIFAR model by using ESH? You stopped learning at epoch 100 and only compared at that moment.
>
> Two reasons:
> 1. Since we were mostly concerned with the speed of convergence, it is more interesting to see which method is doing better at early epochs.
> 2. The resources required for training EBMs to convergence are steep, which is a primary motivation for this work. We ran out of time before submission. We will include some results for later epochs in the final version, though the same qualitative result holds: ESH sampling converges faster.

---

> > ### Author Response · Authors · 2021-08-09
> > **Normalization clarifications**
> >
> > We want to give a more detailed response to this question:
> >
> > > For line 86, what happens if the support of p(x) is unbounded, e.g. R^d? We cannot define a uniform distribution here, and in your formulation the identity in (3) seems crucial. Even then would your method be established?
> >
> > We have two distributions of interest, the joint distribution, $p(x,v) = \delta(E(x) + K(v) - c) / Z'$ and the Gibbs distribution on the marginal, $p(x) = e^{-E(x)} / Z$.
> >
> > It doesn't matter if the support of $x$ or $v$ is unbounded, as long as the energy function, $E(x)$ is normalizable. If $x \in \mathbb R^d$ but $E(x) = x^2$, then $Z = \int dx ~e^{-x^2}$ will be finite. The assumption that the energy function is normalizable is a common one, and we will make that explicit in the final version.
> >
> > In case the reviewer is actually interested in whether $Z'$ is finite, we note that our proof implies that $Z$ is proportional to $Z'$. So if $Z$ is finite, $Z'$ will be also.

---

> > ### Comment · Reviewer_T1qm · 2021-08-14
> > **Are these right?**
> >
> > Thank you for clarification. I want to know if I understood correctly:
> >
> > You have Algorithm 1 (leapfrog, or ESH-leap), Algorithms 2 (ergodic) and 3 (Jarzynski). And, though not named as an algorithm, you have ESH-RK, ESH-RK-orig, and ESH-leap-orig.
> >
> > Algorithm 1 is itself a sampler, and is wrapped by Algorithm 2 and Algorithm 3. And the experiment in 3.1 compares Algorithm 1 with  ESH-RK, ESH-RK-orig, ESH-leap-orig, but no test on Algorithm 2 and 3. And the experiments in 3.2, 3.3 compare Algorithm 1 with other MCMC methods (Isn’t it of CIFAR? You answered that for all the CIFAR experiments the Jarzynski sampler was used, which I guess as Alg. 3. But, in 258, you mention that ESH-leap is used…??). Then, in experiment 3.4, Algorithm 3 is used.
> >
> > I was not aware of the problem of ergodicity and stochasticity that reviewers nN2s and euC9 pointed out. Now you emphasize that if we use Algorithm 3, ergodicity does not become a problem.
> >
> > Are these right?

---

> > > ### Author Response · Authors · 2021-08-14
> > > **Algorithm clarifications**
> > >
> > > There are two technical parts to the method:
> > > 1) How do we simulate the ESH dynamics (defined by an ODE, Eq. 2)?
> > > 2) Given a numerical integrator for ESH dynamics, how do we construct a sampler?
> > >
> > > We investigate these two questions separately.
> > >
> > > 1) How do we simulate the ESH dynamics?
> > >
> > > ESH-leap-orig (Eq. 5), ESH-leap (Alg. 1 and Eq. 7), ESH-RK, ESH-RK-orig (use RK solver from reference 42) all reference ways to solve the ODE. These are compared in Sec. 3.1 and Fig. 3.  These methods (including Algorithm 1) are *not* samplers, they are numerical integrators for an ODE. We concluded from Sec. 3.1 that ESH-leap is by far the best integrator, so we only use that one in subsequent sections.
> > >
> > > 2) Given a numerical integrator for ESH dynamics, how do we construct a sampler?
> > >
> > > Algorithm 2 and 3 provide two qualitatively different ways to use the ESH dynamics for sampling. Algorithm 2, ergodic sampling, gives unweighted samples so it can be compared to other MCMC samplers in Sec. 3.1 using metrics like MMD and Effective Sample Size in Sec. 3.2. We need datasets with ground truth densities for these metrics, and those are described in lines 179-190.
> > >
> > > Unfortunately, Algorithm 3, Jarzynski sampling, is a weighted sampler so metrics like MMD and ESS cannot be calculated (as far as we know). Therefore, we use an extrinsic task, training an energy-based model on synthetic (Fig. 7 top) and CIFAR (Fig. 7 bottom) data in Section 3.3 and 3.4.
> > >
> > >
> > > > I was not aware of the problem of ergodicity and stochasticity that reviewers nN2s and euC9 pointed out. Now you emphasize that if we use Algorithm 3, ergodicity does not become a problem.
> > >
> > > That's correct. Ergodicity is discussed in Sec. 2.1 "Ergodic Dynamics...".
> > >
> > > The derivation of Alg. 3 in Sec. 2.3 does not require ergodicity, and that is the main point of this section. We will clarify and emphasize this in the final version.
> > >
> > > We will also emphasize the point about stochasticity from Reviewers nN2s and euC9, that the important difference is not that we avoid Metropolis-Hastings rejections (there are many ways to accomplish this), but rather that we avoid stochastic transitions (random walk or momentum randomization in HMC).
> > >
> > > Thank you for your comments, we will also see if we can distinguish the role of the ODE integrator (Alg. 1) and the sampling scheme (Alg. 2&3) more clearly in the final version.

---

### Official Review · Reviewer_daEc · 2021-07-16

**Rating:** 7
**Confidence:** 4

**Summary:**

This paper presents a new method for drawing samples from a Gibbs measure using evolution of a deterministic Hamiltonian ODE. In particular, the authors present their method as a possible alternative to Hamiltonian MCMC. The authors develop an ODE solver that is tailored to their model and present experiments learning synthetic distributions and deep energy based image models.

**Limitations And Societal Impact:**

The authors have adequately address limitations and societal impacts.

**Main Review:**

The EBM learning method presented in this work is very novel and the authors present an interesting ODE alternative to standard EBM learning that is powered by Langevin dynamics/HMC. The central innovation is replacing the standard kinetic energy of HMC (from a probabilistic perspective, the negative log likelihood of a Gaussian) with a different kinetic energy. The derived learning equations give an ODE which, under an assumption of ergodicity, is distributed according to the measure of the learned EBM over chaotic trajectories, allowing the authors to use deterministic instead of stochastic dynamics as is typical. This innovation is quite interesting and a strong aspect of the work.

While the overall idea is strong, there are some areas of this work that could be improved. Some questions and points for discussion:
* Figure 1 is misleading. Langevin sampling should be implemented with a much larger step size for fair comparison, and from my experience with the right step size it should be able to travel to at least a few nearby modes in a few hundred steps. In fact, Langevin sampling can sample fairly from an isotropic gaussian in a single update if epsilon is equal to the gaussian standard deviation. This is very different form what the diagram suggests due to improper tuning in the digram.
* I am a bit confused by the appearance of the data dimension $d$ in the kinetic energy formula. Actually, the inclusion of $d$ appears to lead to a minor issue with the third line of the derivation following equation (3) because I am not sure how the $E(x)/d$ term in the exponential is becomes $E(x)$ in the next step, since this exponential factor can't be absorbed into the normalizer ($d$ is a kind of temperature factor is seems). Is the inclusion of $d$ necessary in the kinetic energy?
* The results for the experiments with synthetic datasets demonstrate the usefulness of the proposed method. The EBM learning experiments are not as convincing. In my experience ULA with 50 steps can achieve good results with correct tuning on both of the tasks presented in Figure 7 and I am not sure this figure is a fair comparison. In the appendix, the authors show that ESH can generate reasonable samples on par with ULA for Cifar10 but the benefit of the new method is unclear. Nonetheless, it is good to see that the proposed method can work at a reasonable scale in practice.
* There is a subtle theoretical complication when using this method to learn an EBM. For standard MCMC-based learning, an update that follows detailed balance ensures that the samples are from the steady state even for short run distributions, as long as the initialization distribution is (approximately) equal to the steady state. While this assumption often does not hold in practical EBM training, it can be ensured to some extent with careful implementation and persistent chains as discussed in [22]. Supposing that you are using ESH to learn an EBM and the ODE initialization distribution is close to the steady state. Does the ergodic hypothesis ensure that short-run evolution under ESH from steady-state initialization has a distribution that still follows the steady state? I am not sure that using ESH to learn an EBM is entirely justified for this reason.

**Overall**: The authors present a novel and interesting deterministic sampling framework for generating samples from unnormalized densities. The experiments show improved results over baseline samplers on synthetic distributions and reasonable results on deep EBMs. There are a few points which could be improved and I am open to changing my score based on author response and revisions.

**Time Spent Reviewing:**

6

---

> ### Author Response · Authors · 2021-08-06
> **Tuning and short-run behavior**
>
> > Figure 1 is misleading. Langevin sampling should be implemented with a much larger step size for fair comparison...
>
> Thank you for bringing this to our attention, you are absolutely correct. We made this figure early on, before we derived the leapfrog integrator, and never revised it. The leapfrog integrator mixes much more quickly than the off-the-shelf Runge-Kutta integrator used in Figure 1, which needed small step sizes for stability and therefore 500 gradient steps to make progress. You can see in Fig. 10 that the ESH leapfrog integrator does good mixing with only 20 gradient evaluations. If we tune the Langevin algorithm (the best step size is 0.1, proportional to the standard deviation of the modes and inter-mode spacing), it mixes very well in 500 gradient evaluations. However, it still does poorly with only 20 gradient evaluations, mixing to one or two neighboring modes at most. We will update Fig. 1 with the best settings for ESH and ULA and also add the version of Fig. 10 that visualizes mixing among competing methods.
>
> > I am a bit confused by the appearance of the data dimension d in the kinetic energy formula... Is the inclusion of d necessary in the kinetic energy?
>
> In $K=d/2 \log v^2/d$, the first $d$ is necessary but the second one is not. The second adds a constant and is only included to show in Appendix A that Newtonian and ESH kinetic energies are on different ends of a q-exponential spectrum. The first $d$ cancels in a tricky way. The volume element gives $\rho^{d-1}$ (in red in the proof), and we get an extra factor of $\rho$ from the delta function change of variables, then the delta function sets $\rho = e^{-E/d}$ so that $\rho^d = (e^{-E/d})^d = e^{-E}$.
>
> > The results for the experiments with synthetic datasets demonstrate the usefulness of the proposed method. The EBM learning experiments are not as convincing. In my experience ULA with 50 steps can achieve good results with correct tuning on both of the tasks presented in Figure 7 and I am not sure this figure is a fair comparison. In the appendix, the authors show that ESH can generate reasonable samples on par with ULA for Cifar10 but the benefit of the new method is unclear. Nonetheless, it is good to see that the proposed method can work at a reasonable scale in practice.
>
> You are right that Fig. 7 is an “unfair” comparison because we purposely don’t do all the special tricks that are required to make ULA converge (like adding a large amount of data noise, or using smaller embedding bottlenecks, see discussion in D.3). ESH does well despite these omissions. In App. D.3 we also show the “fair” comparison, where hyper-parameters are tuned to the best case scenario for ULA (using settings from the thorough study of Nijkamp et al). ULA does produce reasonable samples in a few epochs, but ESH still converges better (at least visually when looking at sampling from random initialization).
>
> > There is a subtle theoretical complication when using this method to learn an EBM. For standard MCMC-based learning, an update that follows detailed balance ensures that the samples are from the steady state even for short run distributions, as long as the initialization distribution is (approximately) equal to the steady state. While this assumption often does not hold in practical EBM training, it can be ensured to some extent with careful implementation and persistent chains as discussed in [22]. Supposing that you are using ESH to learn an EBM and the ODE initialization distribution is close to the steady state. Does the ergodic hypothesis ensure that short-run evolution under ESH from steady-state initialization has a distribution that still follows the steady state? I am not sure that using ESH to learn an EBM is entirely justified for this reason.
>
> This is a nice point, and we’d like to add discussion of this to the paper. Only the Jarzynski formulation tells us anything about short-run behavior. Looking at the Jarzynski formulation in Eq. 8, imagine that the initialization distribution, $q_0$, is actually equal to the target distribution. In that case, the energy terms in “w” cancel out, leaving only the change in log |v|. Due to energy conservation, the mean of this term will be zero with low variance (again, only if $q_0 = p$). Therefore, the weights ($e^w$) will all be peaked at 1, leading to an estimator with low variance, and giving some justification for a heuristic where we simply use unweighted samples. We will develop this point in the appendix.
>
> Edit: We should also point out that it can be shown that if you start with samples from the true Gibbs distribution and run ESH dynamics that you will stay in the same Gibbs distribution. In other words, the Gibbs distribution is invariant under ESH dynamics (MCMC also has this invariance property which justifies sampling short chains when starting from the true distribution). We will add this result in the revised version.

---

> > ### Comment · Reviewer_daEc · 2021-08-13
> > **Thanks for the clarifications**
> >
> > The authors addressed my concerns in their response and I changed my score. One important revision would be to give a visually accurate representation of the relative sampler performances in the Fig. 1. Discussion of the implications of short-run ESH with respect to the Gibbs distribution would also strengthen the theoretical presentation.
> >
> > Edit: I also agree with reviewer euC9 that an investigation of Lyapunov exponents might be helpful to diagnose chaos. To my understanding, the chaotic nature of the sampling paths is an essential part of the ergodicity claim, because it makes the idea of a ergodic measure meaningful. But the authors in their response to euC9 claim that their dynamics might not be chaotic. Can non-chaotic dynamics still have a ergodic measure? Wouldn't this imply either a stationary point or cyclic behavior? I do not think that these points should necessarily be the responsibility of the work at hand to fully develop, but a discussion and numerical investigation of chaotic behavior would improve the clarity of the work.

---

### Official Review · Reviewer_euC9 · 2021-07-16

**Rating:** 8
**Confidence:** 4

**Summary:**

The paper proposes a novel Hamiltonian to define the sampling dynamics in the state space. The main difference with the classical Newtonian Hamiltonian is a different kinetic energy. The proposed kinetic energy is $K(v) = d/2\log(v^2/d)$, where $v$ is the momentum, and $d$ is the dimensionality of the state space. The peculiar property (which is the central point of study) of this Hamiltonian is that the velocity $\dot{x} = v/(v^2/d)$ decreases with the increase of the kinetic energy. Intuitively, this energy defines very reasonable dynamics: the particle slows down in the high-density regions and accelerates with the decrease of the density. Note that the classical HMC behaves oppositely.

For the proposed dynamics, the authors carefully develop technical details. They hypothesize that, unlike HMC, this dynamics could actually be ergodic in the state space and the resampling of the momentum is superfluous then. They also develop a specific integration scheme (based on Leap-Frog) that facilitates an efficient simulation of the dynamics. Throughout the paper, the authors claim that the accept/reject step is not necessary for the correct sampling if the integration is precise enough.

For the empirical evaluation, the authors consider several synthetic distributions and compare the algorithm against similar "physical" samplers. For the synthetic distributions, the authors report ESS and MMD. There is also a brief comparison against other samplers on the energy-based models, where the authors report the average energy of the samples and provide visual examples.

**Limitations And Societal Impact:**

The authors have adequately addressed the limitations and potential negative societal impact of their work.

**Main Review:**

There are several major weak points in the paper. The first one is the ergodicity of the procedure. Indeed, the set $H(x,v) = c$ could be disconnected (say like an elliptic curve). Although the authors demonstrate the convergence to the target empirically, this could be achieved by averaging across many parallel chains, which authors use in the experiments. Also, the authors propose an alternative procedure formulating the particle sampling method (flow interpretation in the paper) based on this dynamics. This one doesn't require the ergodicity assumption but is not tested empirically in the paper (the only empirical evaluation is for the 1-D standard normal in the Appendix).

The second weak point is the simulation without accept/reject step. Indeed, if we simulate the dynamics precisely enough we preserve the target measure and the acceptance test always yields 1. However, this also holds for the HMC algorithm. Since no developments facilitating the removement of the accept/reject step has been made, I don't see any reason to remove it. To be more precise, the authors develop a novel integration scheme, but its development is motivated by the difficulties introduced by the novel kinetic energy. Throughout the paper (line 125, lines 264-265) the authors claim that their development avoids accept/reject, but this could be also avoided for HMC in the same manner. If this could be done, I would expect a comparison with the corresponding modification of HMC.

My final concern is the empirical study. The authors demonstrate that the Hamiltonian is approximately preserved in Fig. 12 of the appendix, but they don't report the step size for these simulations. At the same time, the step size in the experiments is quite agressive (as described by the selection procedure in Section 3.1). So I wonder whether the step size is the same for both experiments? Furthermore, the authors compare the training speed of energy models using different samplers as a subroutine. However, for the comparison, the authors rely on the visual quality of the samples. The pictures look like blobs of different colors, no specific patterns are visible. I don't think that the different colors could be interpreted as a faster convergence.

Minor comments:
- The visualization of the dynamics in Fig. 1 seems a bit weird. Does this width of the "ribbon" appear as fast oscillations of the dynamics?
- line 110. I wouldn't claim that the dynamics is actually chaotic. To classify it as chaotic one needs to estimate the Lyapunov exponent or Kolmogorov-Sinai entropy. Moreover, the proposed algorithm heavily relies on the precise simulation of the dynamics (because of the lack of accepts/rejects); hence, the chaotic property could be very unfavorable from this point of view.

Despite all my concerns, I think the paper is a good fit for NeurIPS. The question of ergodicity of deterministic dynamics seems quite challenging, and I wouldn't expect it to be solved in a single paper. The second issue could be easily solved by the recent developments for non-equilibrium sampling see [1,2,3]. I think this will actually improve the performance of the proposed algorithm. Despite some flaws in the empirical study, I think the paper is important for the MCMC field. It gives a fresh look at Hamiltonian Monte Carlo and poses interesting questions for the community.

I think my final score is a bit biased towards positive because I had the same idea before getting this paper for a review. Nevertheless, I think I wouldn't be the only one excited about this paper in the NeurIPS crowd.

- [1] Rotskoff, Grant M., and Eric Vanden-Eijnden. "Dynamical computation of the density of states and Bayes factors using nonequilibrium importance sampling." Physical review letters 122, no. 15 (2019): 150602.
- [2] Neklyudov, Kirill, and Max Welling. "Orbital MCMC." arXiv preprint arXiv:2010.08047 (2020).
- [3] Thin, Achille, Yazid Janati, Sylvain Le Corff, Charles Ollion, Arnaud Doucet, Alain Durmus, Eric Moulines, and Christian Robert. "Invertible Flow Non Equilibrium sampling." arXiv preprint arXiv:2103.10943 (2021).

**Time Spent Reviewing:**

4

---

> ### Author Response · Authors · 2021-08-06
> **Ergodicity, stochasticity, and chaos**
>
> > There are several major weak points in the paper. The first one is the ergodicity of the procedure...
>
> See *Ergodicity* above.
>
> > The second weak point is the simulation without accept/reject step...
>
> See *Stochasticity* above, this was a really useful point.
>
> > My final concern is the empirical study. The authors demonstrate that the Hamiltonian is approximately preserved in Fig. 12 of the appendix, but they don't report the step size for these simulations. At the same time, the step size in the experiments is quite agressive (as described by the selection procedure in Section 3.1). So I wonder whether the step size is the same for both experiments?
>
> Thanks, we will clarify this. Fig. 12 is from the same experiment shown in Fig. 10 and 11, but we should specify that we are showing the Hamiltonian for the larger of the two step sizes, $\epsilon=0.1$, which is the same choice as discussed in Sec. 3.1. Although the step size seems aggressive, the scaled dynamics have $\dot x = u$ where $u$ is a unit vector. So in some ways that step size is more constrained than in Newtonian dynamics (as in Fig. 5 (right)).
>
> > Furthermore, the authors compare the training speed of energy models using different samplers as a subroutine. However, for the comparison, the authors rely on the visual quality of the samples. The pictures look like blobs of different colors, no specific patterns are visible. I don't think that the different colors could be interpreted as a faster convergence.
>
> As for the EBM training examples, we agree that convergence is hard to quantify. This is why we spent more space on examples where we could quantify convergence using MMD. While we can’t quantify convergence of images, it’s easy to see the instability problem that emerges in EBM training with ULA - the generated images in Fig. 7 for ULA are not even remotely close to correct, even focusing just on the distribution of colors. This experiment demonstrates a key stability problem with EBM training via ULA, even slight changes in hyper-parameters lead to very bad results.
>
> > Minor comments:
> > The visualization of the dynamics in Fig. 1 seems a bit weird. Does this width of the "ribbon" appear as fast oscillations of the dynamics?
>
> The width of the ribbon is proportional to the magnitude of the dynamical velocity, which is proportional to the probability density. So more probable points will have a wider ribbon for these dynamics. The velocity does tend to oscillate in an irregular way as we move in and out of high density modes, but we didn’t include any plots of that.
>
> > line 110. I wouldn't claim that the dynamics is actually chaotic. To classify it as chaotic one needs to estimate the Lyapunov exponent or Kolmogorov-Sinai entropy. Moreover, the proposed algorithm heavily relies on the precise simulation of the dynamics (because of the lack of accepts/rejects); hence, the chaotic property could be very unfavorable from this point of view.
>
> This is a good point, we don’t actually measure the sensitivity to initial conditions so we shouldn’t say that it is chaotic. The key property we need is ergodicity, as discussed above, and we will change the language accordingly. If the dynamics is chaotic, which must depend on the precise energy function sampled, then you raise an interesting point about whether the dynamics can be accurately simulated. This is discussed in the molecular dynamics literature we cited. It is one of the reasons that they prefer Leapfrog integrators which can provide bounds on the error in physical constants, like energy conservation, even if the precise trajectories are chaotic. Since our goal is to ergodically traverse all points of constant energy, and leapfrog integrators can bound the changes in energy, chaotic dynamics may actually be beneficial.
>
> > Despite all my concerns, I think the paper is a good fit for NeurIPS. The question of ergodicity of deterministic dynamics seems quite challenging, and I wouldn't expect it to be solved in a single paper. The second issue could be easily solved by the recent developments for non-equilibrium sampling see [1,2,3]. I think this will actually improve the performance of the proposed algorithm. Despite some flaws in the empirical study, I think the paper is important for the MCMC field. It gives a fresh look at Hamiltonian Monte Carlo and poses interesting questions for the community.
>
> Thank you for the references, they look relevant and we were not familiar with them. We will add them to the final version.

---

> > ### Comment · Reviewer_euC9 · 2021-08-21
> > **after response**
> >
> > I've read the rebuttal and would like to keep my opinion.

---

### Official Review · Reviewer_nN2s · 2021-07-17

**Rating:** 5
**Confidence:** 3

**Summary:**

This paper proposes an approach to sampling from distribution with deterministic Hamiltonian dynamics.

**Ethical Concerns:**

Not applicable.

**Main Review:**

The theory developed and results reported in the paper are not convincing to me:

1. It is unclear how the proposed method can handle the irreducibility issue required by MCMC simulations. To be more detailed, if the proposed sampler returns to the starting point after a short time period, then this trajectory will be repeated for ever due to the nature of its deterministic dynamics.  In this case, the sampler will not be ergodic.

2. The results are reported in graphs in the paper. This makes the reviewers very hard to accurately assess the performance of the proposed method.  The performance of the proposed method should be compared numerically with existing MCMC methods.

**Time Spent Reviewing:**

5

---

> ### Author Response · Authors · 2021-08-06
> **Ergodicity and figures**
>
> > 1. It is unclear how the proposed method can handle the irreducibility issue required by MCMC simulations. To be more detailed, if the proposed sampler returns to the starting point after a short time period, then this trajectory will be repeated for ever due to the nature of its deterministic dynamics. In this case, the sampler will not be ergodic.
>
> See *Ergodicity*, in the common points above.
>
> > 2. The results are reported in graphs in the paper. This makes the reviewers very hard to accurately assess the performance of the proposed method. The performance of the proposed method should be compared numerically with existing MCMC methods.
>
> While we show one set of numerical results in table form in Table 1, the decision of whether to show all the other numerical results from Figs. 3, 4, 6, 8, 9, 11, 12, 13, 14, and 16 as plots or tables may be a matter of taste. If there are raw numbers from some of those plots that you think would be useful to also include in supplementary material in table form, please let us know.

---

> ### Comment · Reviewer_nN2s · 2021-08-31
> **On ergodicity**
>
> I read the rebuttal and feel that the ergodicity issue is not adequately addressed.

---

> > ### Author Response · Authors · 2021-09-01
> > **Better indicators of ergodicity**
> >
> > Thank you for your feedback. For the purpose of improving the manuscript, could you clarify which parts of the ergodicity argument you find unsatisfying?
> >
> > To summarize, our position is that theoretical methods to prove ergodicity are not available even for simple energy models, much less for complex neural network models, and therefore we must focus on empirical evidence. If you feel this is incorrect, we would appreciate some pointers.
> >
> > For empirical measures of ergodicity, there are three ways we propose to strengthen this point in the manuscript.
> > - Effective sample size depends on autocorrelation. The fact that ESH has the largest ESS and smallest autocorrelation is strong evidence against cyclic dynamics. We will add this point to the text.
> > - We will give additional visualization of ESH trajectories, as in Figure 3, but simulated for a single chain for very long times. We will also include a 2-d phase space plot for this trajectory. This will provide visual evidence that even for 2-d densities, the dynamics are not ergodic.
> > - Finally, we will include sampling metrics (our distributional metric, MMD) using samples from a single trajectory over time. If the dynamics are constrained to some subspace (i.e. not ergodic), then this metric will not converge.
> >
> > If none of these empirical indicators seem sufficient, we would be grateful to hear ideas for how we could strengthen this case. Thank you again for taking time to review.

---

### Author Response · Authors · 2021-08-06
**Thanks and common points**

First of all, we thank the reviewers for taking the time to review our paper and give valuable feedback. We are happy to see strong interest in new ways to solve the fundamental problem of sampling.

# Ergodicity

We emphasize the importance of the ergodicity assumption in the paper, and reviewer nN2s and euC9 both raise the possibility of cycles where this could be an issue. The experiment on effective sample size largely addresses this concern. If the dynamics were cyclic, then we would see non-zero autocorrelation of the chains over time. Instead, we see in Table 1, that our effective sample size is much higher than all other methods, because the chains have significantly less auto-correlation. We will include a visualization of this point by including a figure like Fig. 10 for a single very long chain, which makes it clear that the dynamics are ergodic, and accompanying it with the MMD metrics (like the ones in Fig. 3, 4, and 11) but for a single chain sampled over time, to quantify convergence to the true distribution.

Secondly, we emphasize the main reason for including the Jarzynski sampling method is that it shows that the approach can provide unbiased estimators even without assuming ergodicity. In practice, running many chains in parallel is much more typical than running one chain for a long time, which is where non-ergodicity would be particularly detrimental.

Finally, we believe a general theoretical answer to the century old question of which dynamics will or will not be ergodic is unlikely to appear without a major breakthrough. As Rev. euC9 says:
> The question of ergodicity of deterministic dynamics seems quite challenging, and I wouldn't expect it to be solved in a single paper.

Therefore, for the moment it is necessary to rely on empirical probes of ergodicity, like the ones we discussed above.

# Stochasticity

Reviewer euC9 makes this point nicely, HMC with small step size need never reject, and this is the same reason that people use Unadjusted Langevin instead of Metropolis Adjusted Langevin for small step size. Therefore, it doesn’t make sense to say that this is a differentiating factor for ESH, and we agree. MCMC has two stochastic parts: a stochastic transition (e.g. momentum randomization or random walk) and the accept/reject step. We need to clarify that the important part of MCMC that we avoid is the random walk component altogether. This is the key differentiating factor for ESH, since many methods like HMC are designed with the idea of minimizing rejections in mind.

---

### Decision · Program_Chairs · 2021-09-27

**Decision:**

Accept (Poster)

**Comment:**

The paper proposes a sampling method based on deterministic dynamics based on a Hamiltonian with a non-standard kinetic energy. While the referees find the theory part of the manuscript under developed, in particular regarding ergodicity, they agree on the novelty of the method and find the numerical experiments convincing. In view of encouraging novel methods development, while the paper is on the boarderline, the meta-reviewer recommends acceptance as a poster.